# BILEVEL ZOFO: BRIDGING PARAMETER-EFFICIENT AND ZEROTH-ORDER TECHNIQUES FOR EFFICIENT LLM FINE-TUNING AND META-TRAINING

## ABSTRACT

Fine-tuning pre-trained Large Language Models (LLMs) for downstream tasks using First-Order (FO) optimizers presents significant computational challenges. Parameter-Efficient Fine-Tuning (PEFT) methods have been proposed to address these challenges by freezing most model parameters and training only a small subset. While PEFT is efficient, it may not outperform full fine-tuning when high task-specific performance is required. Zeroth-Order (ZO) methods offer an alternative for fine-tuning the entire pre-trained model by approximating gradients using only the forward pass, thus eliminating the computational burden of back-propagation in first-order methods. However, when implementing ZO methods, it is crucial to ensure prompt-based text alignment, and relying on simple, fixed hard prompts may not be optimal. In this paper, we propose a bilevel optimization framework that complements ZO methods with PEFT to mitigate sensitivity to hard prompts while efficiently and effectively fine-tuning LLMs. Our Bilevel ZOFO (Zeroth-Order-First-Order) method employs a double-loop optimization strategy, where only the gradient of the PEFT model and the forward pass of the base model are required. We provide convergence guarantees for Bilevel ZOFO. Empirically, we demonstrate that Bilevel ZOFO outperforms both PEFT and ZO methods in single-task settings. Additionally, we show its strong potential for multitask learning. Compared to current first-order meta-training algorithms for multitask learning, our method has significantly lower computational demands while maintaining or improving performance.

## 1 INTRODUCTION

Fine-tuning pretrained Large Language Models (LLMs) has become a standard approach for down-stream tasks. Traditional first-order (FO) optimizers like Adam (Kingma & Ba, 2015), commonly used for this process, rely on backpropagation. However, as highlighted in Malladi et al. (2023), computing gradients for LLMs can require up to 12 times the memory needed for inference. This scal-ing challenge becomes even more pronounced as models grow larger, imposing significant memory demands and complicating the fine-tuning process, especially in resource-constrained environments.

To address these computational challenges, Parameter-Efficient Fine-Tuning (PEFT) methods have been developed. These techniques freeze most of the model's parameters and train only a small subset, significantly reducing both memory and computational overhead. Popular PEFT approaches include prompt tuning, LoRA fine-tuning, and prefix tuning. Prompt tuning (Lester et al., 2021; Qin & Eisner, 2021; Yu et al., 2023) optimizes continuous prompt vectors that are concatenated with the input embeddings, while prefix tuning (Li & Liang, 2021) introduces learnable prefix tokens that serve as conditioning variables at each transformer layer. LoRA (Low-Rank Adaptation) (Hu et al., 2022; Houlsby et al., 2019) modifies the model's attention and feedforward layers by injecting low-rank trainable matrices, further reducing the resources required for fine-tuning.

While Parameter-Efficient Fine-Tuning (PEFT) methods reduce training costs and memory usage, they may not always achieve the same level of task-specific performance as full model fine-tuning. Research has shown that for tasks requiring high accuracy, complex adaptations, or domain-specific knowledge, full fine-tuning often outperforms PEFT approaches due to its ability to adjust all model

parameters for better adaptation (Hu et al., 2022; Li & Liang, 2021; Zaken et al., 2022). To make full model fine-tuning more computationally feasible, zeroth-order methods offer an alternative by reducing the high computational cost. Rather than computing gradients via backpropagation, zeroth-order methods estimate the gradient using only the forward pass. Initially explored in the 1990s (Spall, 1992; Nesterov & Spokoiny, 2017; Ghadimi & Lan, 2013; Duchi et al., 2015; Liu et al., 2020), these methods have recently gained traction for fine-tuning LLMs (Malladi et al., 2023; Deng et al., 2023a; Ling et al., 2024) and have been shown to be able to outperform first-order PEFT methods given enough training time (Zhang et al., 2024b).

However, zeroth-order methods often rely on simple, fixed prompts during fine-tuning. In tasks like sentiment analysis with the SST-2 dataset, templated prompts (e.g., *"$< CLS >$ text data. It was [terrible | great]. $< SEP >$"*) are crucial for success (Zhang et al., 2024b). These prompts effectively align the text data with task-specific objectives. Therefore, selecting such templates becomes a key hyperparameter, raising the question: Can we automatically discover effective prompts for zeroth-order fine-tuning through prompt tuning? More broadly, can PEFT methods complement zeroth-order fine-tuning for large models? In this work, we propose a new framework to answer this question.

While our focus has thus far been on single-task fine-tuning, many scenarios necessitate multi-task fine-tuning. Multi-task learning (MTL) enables a model to handle multiple tasks simultaneously, fostering knowledge transfer between tasks and improving overall efficiency (Min et al., 2022; Yang et al., 2024). This approach is particularly valuable in low-resource settings, where collecting large labeled datasets can be costly, as is often the case with medical data. In such environments, few-shot learning—where a model is fine-tuned on a high-resource dataset to quickly adapt to new tasks with minimal data—becomes essential (Ye et al., 2021).

To address the challenges of multi-task and few-shot learning in natural language processing, several meta fine-tuning methods have been proposed (Huang et al., 2023; Zhao et al., 2024; Ye et al., 2021; Asadi et al., 2024). However, traditional meta fine-tuning approaches, such as MetaICL Min et al. (2022), still require full-model first-order gradient calculations, which become computationally expensive with large language models (LLMs) containing billions of parameters. Given the success of zeroth-order methods in fine-tuning LLMs for individual tasks, the potential for adapting their applicability to multi-task learning remains largely unexplored.

With the effectiveness of zeroth-order fine-tuning and the advantages of PEFT for single tasks, a natural question arises: Can we develop a new multi-task and few-shot learning methodology that significantly reduces computational costs while maintaining or even enhancing performance? Specifically, can we leverage the efficiency of zeroth-order fine-tuning alongside the adaptability of PEFT within multi-task and few-shot learning for large language models?

## 1.1 CONTRIBUTIONS

In this work, we propose a bilevel framework that leverages Parameter-Efficient Fine-Tuning (PEFT) methods to automatically enhance the performance of zeroth-order fine-tuning for large pre-trained language models. The framework introduces two optimization levels: an upper-level problem focused on fine-tuning the pre-trained base model and a lower-level problem dedicated to selecting the most effective PEFT model for fine-tuning the base model. This dual-level approach allows us to identify the optimal combination of PEFT model and pre-trained model.

To solve the bilevel optimization problem, we propose the Bilevel Zeroth-Order-First-Order (Bilevel ZOFO) method. By incorporating zeroth-order approximations into the first-order bilevel method, Bilevel ZOFO avoids calculating the gradient of the full pre-trained model. Our method further addresses the high memory and computational costs of existing bilevel optimization methods, making it especially suitable for fine-tuning large language models (LLMs) with billions of parameters. Additionally, we provide theoretical guarantees for the convergence of the Bilevel ZOFO method.

Furthermore, we extend our method from single-task to multi-task learning. The zeroth-order fine-tuning at our upper level for the full model significantly reduces the computational cost required compared to existing multi-task learning techniques. Additionally, the use of a PEFT model at the lower level allows for efficient fine-tuning across multiple tasks. The proposed framework, combined

with the newly introduced method, offers an extremely lightweight meta-training process that can be rapidly adapted to new tasks.

We conducted extensive experiments to verify the effectiveness of Bilevel ZOFO. In single-task settings, the Bilevel ZOFO method outperforms both traditional PEFT and standard zeroth-order methods fine tuning on average. In multi-task learning settings, we also show that our method achieves superior results over existing methods.

Overall, our contributions include:

- A bilevel optimization framework that enables zeroth-order fine-tuning for large pre-trained models using PEFT methods for single tasks.

- The Bilevel ZOFO method that is suitable for fine-tuning large pre-trained models and significantly reduces the computational cost of existing bilevel methods, with theoretical convergence guarantees.

- An extremely lightweight meta-training process for multi-task learning.

- Empirical results that validate the superiority of our approach in both single-task and multi-task scenarios.

## 2 RELATED WORK

### 2.1 ZEROTH ORDER IN FINE TUNING LLMS

MeZO (Malladi et al., 2023) is the first work to apply ZO tuning to LLMs. MeZO apply the zeroth-order method to fine-tune large language models (LLMs) for downstream tasks. They demonstrate that their method is compatible with both full-parameter tuning and parameter-efficient tuning techniques, such as LoRA and prefix tuning, while being significantly more computationally efficient. Zhang et al. (2024b) provide a benchmark for zeroth-order optimization in the context of LLM fine-tuning, comparing different zeroth-order optimizers and applying the method to various models. Gautam et al. (2024) introduce variance reduction techniques into zeroth-order methods for fine-tuning, improving both stability and convergence. In addition, zeroth-order methods are applied in federated fine-tuning by Qin et al. (2024) and Ling et al. (2024). Deng et al. (2023b) implement zeroth-order optimization for softmax units in LLMs. Guo et al. (2024b) and Liu et al. (2024b) explore fine-tuning a minimal subset of LLM parameters using zeroth-order methods by sparsifying gradient approximation or the perturbations used in gradient estimation. Tang et al. (2024) investigate the privacy of zeroth-order optimization methods.

In contrast to previous approaches, we propose a bilevel training algorithm that effectively combines the strengths of both first-order parameter-efficient fine-tuning (PEFT) and zeroth-order full-model fine-tuning. Our experiments demonstrate that the bilevel structure, when paired with the most suitable PEFT technique, outperforms both zeroth-order full-model fine-tuning and first-order PEFT methods individually.

### 2.2 FINE-TUNING LLMS FOR MULTITASK AND FEW-SHOT LEARNING

Typical meta-tuning approaches employ first-order methods to train autoregressive LLMs on a multitask dataset for various tasks (Zhong et al., 2021; Min et al., 2022; Guo et al., 2024a). Zhong et al. (2021) apply meta-training to tasks such as hate speech detection, question categorization, topic classification, and sentiment classification. Guo et al. (2024a) adopt the method from Min et al. (2022) for generating stylistic text. While Min et al. (2022) focus on enhancing the in-context learning ability of the meta-trained model for multitask learning, Zhong et al. (2021) focus on improving zero-shot performance.

During training, Min et al. (2022) sample a task from the dataset for each iteration to perform in-context learning. In contrast to Zhong et al. (2021) and Min et al. (2022), our approach uses a bilevel structure: the full LLM is fine-tuned at the upper level, while parameter-efficient fine-tuning (PEFT) models are tuned at the lower level. At test time, we freeze the meta-tuned base model and fine-tune only the PEFT model using a few-shot setup, which is both more cost-effective and efficient. Crucially, we employ a zeroth-order method in meta-tuning the base model at the upper

level, which allows us to bypass the need for backpropagation in the meta-model, significantly reducing computational costs.

## 2.3 PENALTY METHODS FOR BILEVEL OPTIMIZATION

Solving a bilevel optimization problem is challenging because the function value in the upper-level objective depends on the optimizer of the lower-level problem. This makes it difficult to compute the gradient of the upper-level objective, also known as the hypergradient. Classical methods require calculating Hessian-vector multiplications to approximate the hypergradient (Franceschi et al., 2017; 2018; Finn et al., 2017; Li et al., 2022; Rajeswaran et al., 2019; Ghadimi & Wang, 2018; Chen et al., 2022; Lorraine et al., 2020). However, when fine-tuning large language models, this process becomes extremely expensive due to the high computational and memory demands.

Recently, new frameworks for bilevel optimization have been introduced (Lu & Mei, 2024; Shen & Chen, 2023; Liu et al., 2024a; Kwon et al., 2023; Liu et al., 2022). These methods bypass the need for second-order information by reformulating the bilevel problem as a constrained optimization problem. The constraint is penalized, allowing the problem to be tackled as a minimax problem using only first-order information. These methods significantly reduce computational costs by eliminating the need for second-order information. Nevertheless, when fine tuning LLMs, back propagation for calculating the gradient of an LLM is still too expensive.

Liu et al. (2024a) and Lu & Mei (2024) explore the convergence of their proposed methods to the original bilevel problem, while other approaches only demonstrate convergence to the penalized problem. In this paper, we adapt the method from Lu & Mei (2024) to approximate part of the upper-level parameters using a zeroth-order approximation, addressing the challenge posed by the vast number of training parameters in large language models. We also provide convergence guarantees for this adapted zeroth-order-first-order method.

## 3 BILEVEL MODEL AND ZEROTH-ORDER-FIRST-ORDER METHOD

In this section, we present our bilevel model and the zeroth-order-first-order method for solving it. Let $\mathbf{p}$ represent the parameters of the PEFT model, and $\boldsymbol{\theta}$ represent the parameters of the pretrained base model. Given a single downstream task, such as classification, we aim to solve the following optimization problem:

$$\min_{\boldsymbol{\theta} \in \mathbb{R}^d} F(\boldsymbol{\theta}, \mathbf{p}; \mathfrak{D}_f), \tag{1}$$

where $\mathbf{p} \in \mathbb{R}^{d'}$ and $F(\boldsymbol{\theta}, \mathbf{p}; \mathcal{B}) := \frac{1}{|\mathcal{B}|} \sum_{x \in \mathcal{B}} F(\boldsymbol{\theta}, \mathbf{p}; x)$ is a loss function given a dataset $\mathcal{B}$.

When $\mathbf{p}$ corresponds to the embeddings of the hard prompt (as shown in Table 13 in the appendix of Malladi et al. (2023)), the model above reduces to classical fine-tuning on a single downstream task. In model (1), the parameters of the PEFT model, $\mathbf{p}$, are fixed.

To enhance fine-tuning performance, we split the dataset into two parts: one for tuning the PEFT model (denoted as $\mathfrak{D}_{\mathbf{p}}$) and another for fine-tuning the LLM (denoted as $\mathfrak{D}_f$). To maximize performance on downstream tasks, we need the optimal PEFT model parameters that are best suited for the current LLM base model. To achieve this, $\mathbf{p}$ should satisfy the following condition:

$$\mathbf{p}(\boldsymbol{\theta}) \in \arg\min_{\mathbf{p} \in \mathbb{R}^{d'}} F(\boldsymbol{\theta}, \mathbf{p}; \mathfrak{D}_{\mathbf{p}}).$$

This condition reveals that as the parameters $\theta$ of the LLM change, the parameters $\mathbf{p}$ in the PEFT model should also be updated accordingly. Therefore, instead of solving (1), our true objective becomes:

$$\min_{\boldsymbol{\theta} \in \mathbb{R}^d} F(\boldsymbol{\theta}, \mathbf{p}(\boldsymbol{\theta}); \mathfrak{D}_f)$$
$$\text{s.t. } \mathbf{p}(\boldsymbol{\theta}) \in \arg\min_{\mathbf{s} \in \mathbb{R}^{d'}} F(\boldsymbol{\theta}, \mathbf{s}; \mathfrak{D}_{\mathbf{p}}). \tag{2}$$

In this way, we find the optimal pair of parameters for both the PEFT model and the LLM base model to achieve the best performance on downstream tasks.

To solve the bilevel optimization problem (2), classical bilevel methods (as discussed in related work) view (2) as a single-level problem $\min_{\boldsymbol{\theta}} F(\boldsymbol{\theta}, \mathbf{p}(\boldsymbol{\theta}))$. Since $\mathbf{p}(\boldsymbol{\theta})$ is the minimizer of another optimization problem, these methods typically require computing the Hessian-vector product (matrix multiplication of $\nabla_{\boldsymbol{\theta}\mathbf{p}} F(\boldsymbol{\theta}, \mathbf{p})$ and some vector $v$) multiple times to estimate the gradient of $F(\boldsymbol{\theta}, \mathbf{p}(\boldsymbol{\theta}))$ with respect to $\boldsymbol{\theta}$. However, for large language models (LLMs), this approach is computationally prohibitive because the number of parameters in $\boldsymbol{\theta}$ is too large.

To reduce the computational cost, we consider using a penalty method for the bilevel problem (2), as mentioned in related work. Specifically, (2) is equivalent to the following constrained optimization problem:

$$\min_{\boldsymbol{\theta} \in \mathbb{R}^d, \mathbf{p} \in \mathbb{R}^{d'}} F(\boldsymbol{\theta}, \mathbf{p}; \mathfrak{D}_f)$$
$$\text{s.t. } F(\boldsymbol{\theta}, \mathbf{p}; \mathfrak{D}_{\mathbf{p}}) - \inf_{\mathbf{s}} F(\boldsymbol{\theta}, \mathbf{s}; \mathfrak{D}_{\mathbf{p}}) \leq 0. \tag{3}$$

By penalizing the constraint, we obtain the following penalized problem:

$$\min_{\boldsymbol{\theta} \in \mathbb{R}^d, \mathbf{p} \in \mathbb{R}^{d'}} F(\boldsymbol{\theta}, \mathbf{p}(\boldsymbol{\theta}); \mathfrak{D}_f) + \lambda(F(\boldsymbol{\theta}, \mathbf{p}; \mathfrak{D}_{\mathbf{p}}) - \inf_{\mathbf{s} \in \mathbb{R}^{d'}} F(\boldsymbol{\theta}, \mathbf{s}; \mathfrak{D}_{\mathbf{p}})), \tag{4}$$

where $\lambda > 0$. As $\lambda$ increases, the solution to the penalized problem approaches the solution to (3), and thus the solution to (2) (see Lemma 1 for an explicit relationship between the stationary points of (4) and those of the original problem (2)). Note that the penalized problem (4) is equivalent to the following minimax problem:

$$\min_{\boldsymbol{\theta} \in \mathbb{R}^d, \mathbf{p} \in \mathbb{R}^{d'}} \max_{\mathbf{s} \in \mathbb{R}^{d'}} G_\lambda(\boldsymbol{\theta}, \mathbf{p}, \mathbf{s}) := F(\boldsymbol{\theta}, \mathbf{p}(\boldsymbol{\theta}); \mathfrak{D}_f) + \lambda(F(\boldsymbol{\theta}, \mathbf{p}; \mathfrak{D}_{\mathbf{p}}) - F(\boldsymbol{\theta}, \mathbf{s}; \mathfrak{D}_{\mathbf{p}})). \tag{5}$$

In this way, we can solve the bilevel problem as a minimax problem. The basic minimax algorithm works as follows: at iteration $k$, we first solve the maximization problem $\max_{\mathbf{s}} G_\lambda(\boldsymbol{\theta}^k, \mathbf{p}^k, \mathbf{s})$ with $(\boldsymbol{\theta}^k, \mathbf{p}^k)$ fixed. For example, we can update $\mathbf{s}^k$ using an inner loop with stochastic gradient descent (SGD). Let $\mathbf{s}^{k+1}$ be the result of this inner loop. Then, in the outer loop, we update $(\boldsymbol{\theta}^k, \mathbf{p}^k)$ by solving $\min_{\boldsymbol{\theta}, \mathbf{p}} G_\lambda(\boldsymbol{\theta}, \mathbf{p}, \mathbf{s}^{k+1})$ with $\mathbf{s}^{k+1}$ fixed. Again, SGD can be used to update $\boldsymbol{\theta}^k$ and $\mathbf{p}^k$. The conceptual algorithm and pipeline is presented in Algorithm 1 and Figure 3 respectively.

---

**Algorithm 1** Bilevel first-order method

1: Input: $\eta > 0, \zeta > 0, \mathbf{s}^0 = \mathbf{s}^k, K, T \in \mathbb{N}_+, \lambda \geq 0$.
2: **for** k=0,...,K **do**
3:    **for** $t = 0, \ldots, T-1$. **do**
4:       Let $\mathbf{s}_{t+1}^k = \mathbf{s}_t^k - \eta \nabla_{\mathbf{s}} G_{\lambda_k}(\boldsymbol{\theta}^k, \mathbf{p}^k, \mathbf{s}_t^k)$.
5:       Output $\mathbf{s}^{k+1} = \mathbf{s}_T^k$.
6:    **end for**
7:    Let $\boldsymbol{\theta}^{k+1} = \boldsymbol{\theta}^k - \zeta \nabla_{\boldsymbol{\theta}} G_{\lambda_k}(\boldsymbol{\theta}^k, \mathbf{p}^k, \mathbf{s}^k)$ and $\mathbf{p}^{k+1} = \mathbf{p}^k - \zeta \nabla_{\mathbf{p}} G_\lambda(\boldsymbol{\theta}^k, \mathbf{p}^k, \mathbf{s}^k)$.
8: **end for**

---

However, note that

$$\nabla_{\boldsymbol{\theta}} G_{\lambda_k}(\boldsymbol{\theta}^k, \mathbf{p}^k, \mathbf{s}^k) = \nabla_{\boldsymbol{\theta}} F(\boldsymbol{\theta}^k, \mathbf{p}^k; \mathfrak{D}_f) + \lambda_k(\nabla_{\boldsymbol{\theta}} F(\boldsymbol{\theta}, \mathbf{p}^k; \mathfrak{D}_{\mathbf{p}}) + \nabla_{\boldsymbol{\theta}} F(\boldsymbol{\theta}^k, \mathbf{s}^k; \mathfrak{D}_{\mathbf{p}})), \tag{6}$$

requires calculating the gradient with respect to $\boldsymbol{\theta}$, i.e, $\nabla_{\boldsymbol{\theta}} F(\boldsymbol{\theta}^k, \mathbf{p}^k; \mathfrak{D}_f)$. Given the large scale of $\boldsymbol{\theta}$ in LLMs, this is computationally expensive. To avoid this, we use zeroth-order (ZO) information to approximate the gradient $\nabla_{\boldsymbol{\theta}} G$. Following Malladi et al. (2023); Zhang et al. (2024b); Guo et al. (2024b), we employ the Simultaneous Perturbation Stochastic Approximation (SPSA) as a classical zeroth-order gradient estimator. Specifically, at each iteration $k$, we sample $\mathbf{z}^k \sim N(0, I_d)$, where $d$ is the dimension of $\boldsymbol{\theta}$. We then approximate the gradient as follows:

$$\hat{\nabla}_{\boldsymbol{\theta}} F(\boldsymbol{\theta}^k, \mathbf{p}^k; x) := \frac{F(\boldsymbol{\theta}^k + \epsilon \mathbf{z}^k, \mathbf{p}^k; x) - F(\boldsymbol{\theta}^k - \epsilon \mathbf{z}^k, \mathbf{p}^k; x)}{2\epsilon} \mathbf{z}^k. \tag{7}$$

As opposed to the number of LLM parameters $\boldsymbol{\theta}$, the number of PEFT parameters $\mathbf{p}$ is very small. So it is feasible to compute the exact gradient with respect to $\mathbf{p}$. Thus, we calculate $\nabla_{\mathbf{p}} F(\boldsymbol{\theta}, \mathbf{p}; \mathcal{B})$ exactly.

Additionally, in each iteration $k$ and inner iteration $t$ of Algorithm 1, we sample a mini-batch $\mathcal{B}$ of size $B$. We use $\hat{\nabla}_{\boldsymbol{\theta}} F(\boldsymbol{\theta}^k, \mathbf{p}^k; \mathcal{B})$ to substitute $\nabla_{\boldsymbol{\theta}} G_{\lambda_k}(\boldsymbol{\theta}^k, \mathbf{p}^k, \mathbf{s}^k)$ in (6). We also use mini-batches when calculating the gradients with respect to the PEFT parameters **s** and **p**.

This approach leads to the final algorithm (Algorithm 2) for fine-tuning LLMs using the bilevel model (2). We refer to this method as the Bilevel Zeroth-Order-First-Order (Bilevel ZOFO) method.

---

**Algorithm 2** Bilevel Zeroth-order-first-order Method (Bilevel ZOFO)

---

1: Input: $\eta > 0$, $\zeta > 0$, batchsize $B$, $\mathbf{s}^0 = \mathbf{s}^k$, $K, T \in \mathbb{N}_+$, $\lambda > 0$.
2: **for** k=0,...,K **do**
3:     **for** t=0,...,T-1 **do**
4:         Sample a batch $\mathcal{B}_{t,\mathbf{p}}^k$ from $\mathfrak{D}_{\mathbf{p}}$.
5:         Let $\mathbf{s}_{t+1}^k = \mathbf{s}_t^k - \eta \nabla_{\mathbf{s}} F(\boldsymbol{\theta}^k, \mathbf{s}_t^k; \mathcal{B}_{t,\mathbf{p}}^k)$
6:         Output $\mathbf{s}^{k+1} = \mathbf{s}_T^k$.
7:     **end for**
8:     Sample a batch $\{\mathcal{B}_f^k\}$ from $\mathfrak{D}_f$ and $\{\mathcal{B}_{\mathbf{p}}^k\}$ from $\mathfrak{D}_{\mathbf{p}}$.
9:     For $x \in \mathcal{B}_{\mathbf{p}}^k \cup \mathcal{B}_f^k$, calculate $\hat{\nabla}_{\boldsymbol{\theta}} F(\boldsymbol{\theta}^k, \mathbf{p}^k; x)$ following (7).
10:    Let

$$
\begin{aligned}
\mathbf{p}^{k+1} &= \mathbf{p}^k - \zeta \left( \nabla_{\mathbf{p}} F(\boldsymbol{\theta}^k, \mathbf{p}^k; \mathcal{B}_f^k) + \lambda_k \left( \nabla_{\mathbf{p}} F(\boldsymbol{\theta}^k, \mathbf{p}^k; \mathcal{B}_{\mathbf{p}}^k) \right) \right) \\
\boldsymbol{\theta}^{k+1} &= \boldsymbol{\theta}^k - \zeta \left( \hat{\nabla}_{\boldsymbol{\theta}} F(\boldsymbol{\theta}^k, \mathbf{p}^k; \mathcal{B}_f^k) + \lambda_k (\hat{\nabla}_{\boldsymbol{\theta}} F(\boldsymbol{\theta}^k, \mathbf{p}^k; \mathcal{B}_{\mathbf{p}}^k) - \hat{\nabla}_{\boldsymbol{\theta}} F(\boldsymbol{\theta}^k, \mathbf{s}^{k+1}; \mathcal{B}_{\mathbf{p}}^k)) \right)
\end{aligned}
\tag{8}
$$

11: **end for**

---

## 4 EXPERIMENTS

We conduct extensive experiments on various language models of different scales to demonstrate the effectiveness of bilevel-ZOFO in both single-task and multi-task meta-training settings.

### 4.1 BILEVEL-ZOFO FOR SINGLE-TASK FINE-TUNING

Following MeZO (Malladi et al., 2023), we evaluate our approach on a range of classification and multiple-choice tasks. In this setting, training and testing are conducted on the same task. We employ prompt-tuning (Lester et al., 2021), prefix-tuning (Li & Liang, 2021), and LoRA (Hu et al., 2022) for lower-level training to validate bilevel-ZOFO under different conditions and resource constraints. During each lower-level update, we update only the PEFT parameters, and during the upper-level optimization step, we tune the full model using zeroth-order gradient approximation. We perform 10 lower-level updates between each pair of upper-level updates. For each task, we randomly sample 1000 examples for training, 500 examples for validation, and 1000 examples for testing. We use the Adam optimizer (Kingma & Ba, 2015) and report test accuracy or F1-score.

We compare our method against several baselines, including MeZO for Full Model Fine-tuning, MeZO for PEFT, and First-order PEFT. More specifically, MeZO is replacing the gradient in the model with the approximation (7) and then doing SGD or adam. We fix the total memory budget of each step across bilevel-ZOFO and the baselines. We train zeroth-order methods for 10,000 steps, and first-order methods for 5000 steps. We compare the memory requirements of our method with the baselines in Figure 5, and provide wall-clock analysis in Table 6. For all experimental details, refer to the Appendix B.1.3 and Appendix B.1.4.

Table 1 presents the test metrics when applying bilevel-ZOFO and baselines to fine-tune OPT-1.3B (Zhang et al., 2022) on a downstream task. Table 2 demonstrates the results for Llama2-7b (Touvron et al., 2023). We can make the following observations:

**Bilevel-ZOFO outperforms MeZO on almost all tasks:** With the same memory allocation per training step, bilevel-ZOFO outperforms MeZO, even when trained for half the number of iterations across most tasks.

**Bilevel-ZOFO outperforms FO PEFT on average:**

| Trainer | Mode | BoolQ | CB | Copa | ReCoRD | RTE | SST2 | WIC | WinoGrande | WSC | Average |
|---|---|---|---|---|---|---|---|---|---|---|---|
| MeZO | ft | 0.6927 | 0.7767 | 0.7000 | 0.6980 | 0.6587 | 0.8214 | 0.5543 | 0.5480 | 0.5054 | 0.6617 |
| | lora | 0.6860 | 0.7607 | 0.7200 | 0.7083 | 0.6755 | 0.8501 | 0.5549 | 0.5607 | 0.5570 | 0.6748 |
| | prefix | 0.6573 | 0.7945 | 0.7033 | 0.7047 | 0.6972 | 0.8218 | 0.5622 | 0.5370 | 0.5105 | 0.6654 |
| | prompt | 0.6260 | 0.5821 | 0.7067 | 0.7070 | 0.5415 | 0.7463 | 0.5574 | 0.5556 | 0.4654 | 0.6098 |
| | average | 0.6655 | 0.7285 | 0.7075 | 0.7045 | 0.6432 | 0.8099 | 0.5572 | 0.5503 | 0.5096 | 0.6529 |
| FO | lora | 0.7456 | 0.8512 | 0.7500 | 0.7206 | 0.7292 | 0.9258 | 0.6463 | 0.5806 | 0.6474 | 0.7330 |
| | prefix | 0.7300 | 0.8571 | 0.7167 | 0.7093 | 0.7136 | 0.8133 | 0.5387 | 0.5787 | 0.5705 | 0.6920 |
| | prompt | 0.7150 | 0.7142 | 0.7466 | 0.7163 | 0.6936 | 0.8016 | 0.5386 | 0.5980 | 0.5062 | 0.6700 |
| | average | 0.7302 | 0.8075 | 0.7378 | 0.7154 | 0.7121 | 0.8470 | 0.5745 | **0.5857** | 0.5747 | 0.6977 |
| Ours | lora | 0.7433 | 0.9167 | 0.7400 | 0.7183 | 0.7401 | 0.9331 | 0.6447 | 0.5903 | 0.6428 | 0.7410 |
| | prefix | 0.7340 | 0.8690 | 0.7267 | 0.7140 | 0.7304 | 0.8550 | 0.6317 | 0.5710 | 0.5810 | 0.7125 |
| | prompt | 0.7367 | 0.7679 | 0.7633 | 0.7257 | 0.6867 | 0.8335 | 0.6267 | 0.5900 | 0.5133 | 0.6938 |
| | average | **0.7380** | **0.8512** | **0.7433** | **0.7193** | **0.7191** | **0.8739** | **0.6344** | 0.5838 | **0.5790** | 0.7158 |

Table 1: Single-Task Experiments on OPT-1.3B with 1000 samples. Values correspond to mean across three random seeds. FO: First-Order. FT: full-model fine-tuning. See Table 4 in the Appendix for standard deviation values.

Comparing each FO-PEFT setting with the corresponding bilevel-ZOFO setting, we see that bilevel-ZOFO outperforms the corresponding FO-PEFT methods across most instances and **on average**.

**Bilevel-ZOFO outperforms baselines more significantly in resource-constrained settings:** Figure 1 compares the number of parameters tuned by bilevel-ZOFO and first-order baselines. The number of parameters tuned for prefix tuning and prompt tuning is lower than for LoRA. As shown in Table 1, when fewer parameters are tuned, bilevel-ZOFO demonstrates a larger improvement over first-order methods in tuning FEPT models. Since memory usage and training steps remain the same, bilevel-ZOFO proves to be a more suitable option for fine-tuning LLMs in constrained environments compared to PEFT and MeZO.

**Bilevel-ZOFO generalizes effectively to larger LLMs:** Table 2 compares bilevel-ZOFO with the baselines when fine-tuning Llama2-7b (Touvron et al., 2023) on various classification and

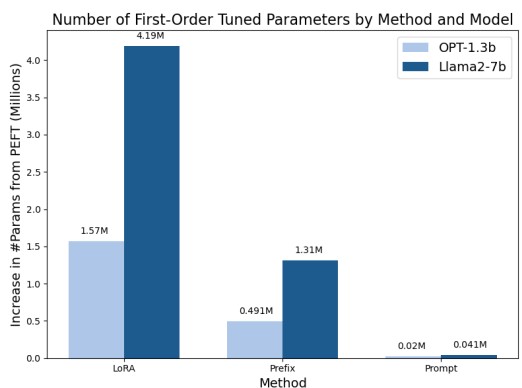

Figure 1: Number of additional parameters PEFT methods introduce to each model.

open-ended generation tasks. The results show that bilevel-ZOFO's advantages are not confined to smaller models like OPT-1.3b, but also extend to larger LLMs.

## 4.2 MULTI-TASK FINE-TUNING EXPERIMENTS

Following the methodology of Min et al. (2022), we evaluate the performance of bilevel-ZOFO as a fast and efficient meta-learning algorithm. We perform experiments using four of the distinct meta-learning settings outlined in MetaICL (Min et al., 2022): classification-to-classification, non-classification-to-classification, QA-to-QA, and non-QA-to-QA. For instance, in non-classification-to-classification setting, we train on a number of non-classification subtasks and test on a number of distinct classification subtasks. Each of these *meta-learning tasks* includes a set of training sub-tasks and a different set of test sub-tasks. The sub-tasks are sourced from CROSSFIT (Ye et al., 2021) and UNIFIEDQA (Khashabi et al., 2020), comprising a total of 142 unique sub-tasks. These sub-tasks cover a variety of problems, including text classification and question answering all in English. We use GPT2-Large Radford et al. (2019) as the base model for these experiments.

We compare our method against several baseline approaches:

| Trainer | Mode | BoolQ | ReCoRD | SQuAD | SST2 | Average |
|---|---|---|---|---|---|---|
| MeZO | ft | 0.7915 | 0.7890 | 0.7737 | 0.8646 | 0.8047 |
| | lora | 0.8020 | 0.7970 | 0.7412 | 0.8529 | 0.7983 |
| | prefix | 0.7830 | 0.7905 | 0.7093 | 0.8364 | 0.7798 |
| | prompt | 0.7787 | 0.7935 | 0.7014 | 0.8246 | 0.7746 |
| | average | 0.7888 | 0.7925 | 0.7489 | 0.8397 | 0.7825 |
| FO | lora | 0.8420 | 0.7920 | 0.8197 | 0.9557 | 0.8524 |
| | prefix | 0.7783 | 0.8013 | 0.7946 | 0.9243 | 0.8246 |
| | prompt | 0.8083 | 0.8023 | 0.7805 | 0.9284 | 0.8299 |
| | average | **0.8095** | 0.7985 | 0.7983 | 0.9361 | 0.8356 |
| Ours | lora | 0.8473 | 0.8290 | 0.8160 | 0.9629 | 0.8638 |
| | prefix | 0.8193 | 0.8067 | 0.8090 | 0.9382 | 0.8433 |
| | prompt | 0.8145 | 0.8108 | 0.7960 | 0.9222 | 0.8359 |
| | average | 0.7937 | **0.8155** | **0.8070** | **0.9414** | 0.8394 |

Table 2: Single-Task Experiments on Llama2-7B with 1000 samples. Values correspond to mean across three random seeds. FO: First-Order. FT: full-model fine-tuning. See Table 5 for full details.

- **MetaICL** (Min et al., 2022): A method for meta-learning with in-context learning. MetaICL tunes all the parameters of the base model using the first-order method. In both training and testing, the model is given $k$ demonstration examples, $(a_1, b_1), \ldots, (a_k, b_k)$, where $b_i$ represents either classification labels or possible answers in question-answering tasks, along with one test example $(a, b)$. The input is formed by concatenating the demonstration examples $a_1, b_1, \ldots, a_k, b_k, a$. The model then computes the conditional probability of each label, and the label with the highest probability is selected as the prediction.

- **Zero-shot**: This method uses the pretrained language model (LM) without any tuning, performing zero-shot inference without any demonstration examples.

- **In-context Learning (ICL)**: This method uses the pretrained LM with in-context learning by conditioning on a concatenation of $k$ demonstration examples and 1 actual test sample similar to MetaICL.

We sample 768 examples from each training sub-task. We train MetaICL in their original setting for 30,000 steps. To train our method, we split the training dataset of each sub-task to two subsets, 256 samples as the development dataset for upper-level updates and 512 samples for lower-level training. For each outer iteration of our method, we randomly sample a subset of 5 training tasks. We perform 10 lower-level updates between each pair of upper-level updates. To keep bilevel-ZOFO as lightweight as possible, unlike MetaICL, we do not include demonstration examples in the inputs. Since bilevel-ZOFO uses significantly less memory and has much faster updates compared to MetaICL, theoretically we are able to train it for many more iterations within the same total training duration as MetaICL. However, due to resource constraints, we only train bilevel-ZOFO for 50,000 iterations. Similar to Malladi et al. (2023), we did not observe a plateau in performance for bilevel-ZOFO, indicating that further training can yield additional improvements.

For both ICL and MetaICL, during the testing phase the model is given $k = 4$ demonstration examples for each test data point. We don't use demonstration examples in test samples for bilevel-ZOFO evaluation. We evaluate the zero-shot capabilities of our method as well as the performance of the final model LoRA-tuned for 10 additional iterations on 4 demonstration samples from each class of each test sub-task. Similar to Min et al. (2022), we report **Macro-averaged F1** as the evaluation metric. See Appendix B.4 for all training details.

Table 3 presents the Meta-learning results. We observe that zero-shot bilevel-ZOFO outperforms zero-shot on all tasks. While bilevel-ZOFO does not surpass ICL or MetaICL in the zero-shot setting, this is expected. It is crucial to note that 1) MetaICL fine-tunes the entire base model using first-order methods, which incurs a significantly higher computational cost. Additionally, as noted by Malladi et al. (2023) and confirmed in our experiments, zeroth-order methods typically require many more iterations to converge, with performance improving as training progresses. 2)Both ICL and MetaICL with $k = 4$ demonstration examples take 4 times more time to do inference than a method with no demonstration examples.

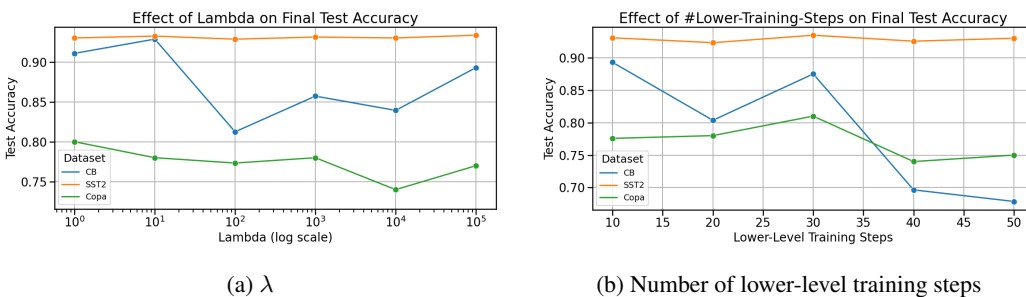

(a) $\lambda$        (b) Number of lower-level training steps

Figure 2: Ablation over $\lambda$ in (5) and the number of lower-level training steps before each upper-level update.

Nonetheless, after a lightweight 10-iteration LoRA fine-tuning phase, bilevel-ZOFO surpasses ICL and MetaICL on nearly every hyper-task, highlighting its strong potential as a meta-learning algorithm.

| Method | class → class | non_class → class | qa → qa | non_qa → qa |
|---|---|---|---|---|
| Zero-shot | 34.2 | 34.2 | 40.2 | 40.2 |
| Few-shot | 34.9 (1.4) | 34.9 (1.4) | 40.5 (0.3) | 40.5 (0.4) |
| MetaICL | 46.4 (1.1) | 37.7 (1.7) | **45.5** (0.3) | 40.2 (0.6) |
| Ours (Zero-shot) | 34.5 | 34.3 | 41.8 | 40.4 |
| Ours(Tuned) | **47.1** | **42.4** | 43.5 (1.3) | **41.9** |

Table 3: Multi-task Meta learning results using GPT2-Large as the base model. Values correspond to the mean and standard deviation over 5 test seeds which include different demonstration samples for each test task. class: Classification, qa: Question Answering

### 4.3 ABLATIVE STUDIES

We perform an ablation study by varying the regularization parameter $\lambda$ (as defined in Equation (5)) and the number of lower-level training steps between each pair of upper-level updates. Figure 2 shows the results. From Figure 2a, the effect $\lambda$ appears to be non-linear, indicating the need to find an optimal balance. Nonetheless, a moderate value like 10 or 100 seems to work reasonably well on all tasks. As anticipated, Figure 2b demonstrates that performance generally degrades when the total number of upper-level updates is reduced, suggesting there is a trade-off between latency and performance. While more upper-level updates improve results, they also extend the overall training time.

## 5 ANALYSIS

In this section we give convergence guarantee for Bilevel ZOFO. Suppose $(\boldsymbol{\theta}, \mathbf{p}) \in \mathbb{R}^{d+d'}$ and $\mathbf{s} \in \mathbb{R}^{d'}$. The following assumptions are made throughout this section.

**Assumption 1.** *We make the following assumptions:*

- *$G(\boldsymbol{\theta}, \mathbf{p}, \cdot)$ can be potentially nonconvex and $G(\cdot, \cdot, \mathbf{s})$ is $\tau-$ strongly concave; $F(\boldsymbol{\theta}, \mathbf{p})$ is twice continuously differentiable in $\boldsymbol{\theta}, \mathbf{p}$.*

- *$G$ is $\ell$-Lipschitz smooth in $\mathbb{R}^{d+2d'}$, i.e. $\forall (\boldsymbol{\theta}_1, \mathbf{p}_1, \mathbf{s}_1), (\boldsymbol{\theta}_2, \mathbf{p}_2, \mathbf{s}_2) \in \mathbb{R}^{d+2d'}$,*

$$\|\nabla G(\boldsymbol{\theta}_1, \mathbf{p}_1, \mathbf{s}_1) - \nabla G(\boldsymbol{\theta}_2, \mathbf{p}_2, \mathbf{s}_2)\| \leq \ell \|(\boldsymbol{\theta}_1, \mathbf{p}_1, \mathbf{s}_1) - (\boldsymbol{\theta}_2, \mathbf{p}_2, \mathbf{s}_2)\|.$$

*We define $\kappa := \ell/\tau$ as the problem condition number.*

- $\forall (\boldsymbol{\theta}, \mathbf{p}, \mathbf{s}) \in \mathbb{R}^{d+2d'}$, *sample estimates satisfy*

$$\mathbb{E}[G(\boldsymbol{\theta}, \mathbf{p}, \mathbf{s}; \xi)] = G(\boldsymbol{\theta}, \mathbf{p}, \mathbf{s}),$$
$$\mathbb{E}[\nabla G(\boldsymbol{\theta}, \mathbf{p}, \mathbf{s}; \xi)] = \nabla G(\boldsymbol{\theta}, \mathbf{p}, \mathbf{s}),$$
$$\mathbb{E}\|\nabla G(\boldsymbol{\theta}, \mathbf{p}, \mathbf{s}; \xi) - \nabla G(\boldsymbol{\theta}, \mathbf{p}, \mathbf{s})\|^2 \le \frac{\sigma^2}{B}$$

  *for sample $\xi$ with size $|\xi| = B$ and constant $\sigma > 0$.*

- $\max_{\mathbf{s}} G(\boldsymbol{\theta}, \mathbf{p}, \mathbf{s})$ *is lower bounded.*

We first discuss the relationship between the optimality condition (4) and (2). We start with defining the $\epsilon$-stationary points of (4) and (2) for general bilevel and minimax problems [1].

**Definition 1.** *Given a bilevel optimization problem*

$$f^* = \min_{\mathbf{x}} f(\mathbf{x}, \mathbf{y}^*(\mathbf{x})), \mathbf{y}^*(\mathbf{x}) \in \arg\min_{\mathbf{z}} g(\mathbf{x}, \mathbf{z})$$

*and any $\epsilon > 0$, a point $(\mathbf{x}_\epsilon, \mathbf{y}_\epsilon)$ is called an $\epsilon$-stationary point if*

$$\mathbb{E}[\|\nabla f(\mathbf{x}_\epsilon, \mathbf{y}^*(\mathbf{x}_\epsilon))\|] \le O(\epsilon), f(\mathbf{x}_\epsilon, \mathbf{y}_\epsilon) - \min_{\mathbf{z}} f(\mathbf{x}_\epsilon, \mathbf{z}) \le \epsilon.$$

**Definition 2.** *Given a minimax problem*

$$f^* = \min_{\mathbf{x}} \max_{\mathbf{y}} f(\mathbf{x}, \mathbf{y})$$

*and any $\epsilon > 0$, a point $(\mathbf{x}_\epsilon, \mathbf{y}_\epsilon)$ is called an $\epsilon$-stationary point if*

$$\mathbb{E}[\|\nabla_{\mathbf{x}} f(\mathbf{x}_\epsilon, \mathbf{y}_\epsilon)\|^2] \le \epsilon^2, \ \mathbb{E}[\|\nabla_{\mathbf{y}} f(\mathbf{x}_\epsilon, \mathbf{y}_\epsilon)\|^2] \le \epsilon^2.$$

**Lemma 1.** *If assumption 1 holds and $\lambda = 1/\epsilon$, assume that $\nabla^2 F(\boldsymbol{\theta}, \cdot)$ is Lipschitz continuous and $(\boldsymbol{\theta}, \mathbf{p}, \mathbf{s})$ is an $\epsilon$-stationary point of (4), then $(\boldsymbol{\theta}, \mathbf{s})$ is an $\epsilon$-stationary point of (2).*

The following is the low effective rank assumption from Malladi et al. (2023). This assumption avoids dimension $d$ in the total complexity. Following Malladi et al. (2023), we assume here that $\mathbf{z}^k$ in (7) is sampled from shpere in $\mathbb{R}^d$ with radius $\sqrt{d}$ for ease of illustration.

**Assumption 2.** *For any $(\boldsymbol{\theta}, \mathbf{p}, \mathbf{s}) \in \mathbb{R}^{d+2d'}$, there exists a matrix $H(\boldsymbol{\theta}, \mathbf{p}, \mathbf{s})$ such that $\nabla^2 G(\boldsymbol{\theta}, \mathbf{p}, \mathbf{s}) \preceq H(\boldsymbol{\theta}, \mathbf{p}, \mathbf{s}) \preceq \ell \cdot I_d$ and $tr(H(\boldsymbol{\theta}, \mathbf{p}, \mathbf{s})) \le r \cdot \|H(\boldsymbol{\theta}, \mathbf{p}, \mathbf{s})\|$.*

**Theorem 1.** *If Assumptions 1 and 2 hold, by setting*

$$\eta = \frac{1}{2\ell}, \zeta = \frac{1}{2\ell r}, \lambda = \frac{1}{\epsilon}, B = O(\sigma^2 \epsilon^{-2}),$$
$$\alpha = O(\epsilon \kappa^{-1} (d + d')^{-1.5}), T = O\left(\kappa \log(\kappa \epsilon^{-1})\right), K = O(\kappa r \epsilon^{-2}),$$

*there exists an iteration in Algorithm 2 that returns an $\epsilon$-stationary point $(\boldsymbol{\theta}, \mathbf{p}, \mathbf{s})$ for (5) and it satisfies*

$$\mathbb{E}[\|\nabla F(\boldsymbol{\theta}, \mathbf{p}^*(\boldsymbol{\theta}); \mathcal{D}_f)\|] \le O(\epsilon), F(\boldsymbol{\theta}, \mathbf{s}; \mathcal{D}_\mathbf{p}) - \min_{\mathbf{p}} F(\boldsymbol{\theta}, \mathbf{p}; \mathcal{D}_\mathbf{p}) \le \epsilon.$$

**Remark 1.** *The total number of zeroth order gradient calculations is*

$$TKB_1 + KB_2 = O(\sigma^2 \kappa^2 r \epsilon^{-4} \log(\kappa \epsilon^{-1})).$$

*This result matches the complexity in previous zeroth order minimax algorithm in Wang et al. (2023) but solves our bilevel optimization problem (2) and does not depend on the dimensionality $d$ thanks to the efficient rank assumption 2, providing efficiency guarantee for our algorithm.*

## 6 CONCLUSIONS

In this work, we introduced a novel bilevel optimization framework designed to mitigate the downsides of PEFT and zeroth-order full model fine-tuning. We propose a new method that is more efficient than existing bilevel methods and thus more suitable for tuning full pre-trained large language models. Theoretically, we provide convergence guarantees for this new method. Empirically, we show that this method outperforms both zeroth-order methods and PEFT methods when solving one single task settings on average. Additionally, we demonstrate that this method is effective and efficient when adapted to do multi-task learning. With competitive and even better performance compared to existing meta-training methods, our method offers a significantly cheaper training process.

---

[1] In the following definitions, the expectation is taken over the randomness in the algorithm that $(\mathbf{x}, \mathbf{y})$ is generated.

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

## A METHOD

### A.1 PROOFS

In the proofs we use the simplified notations $\mathbf{x} := (\boldsymbol{\theta}, \mathbf{p})$, $\mathbf{y} := \mathbf{s}$, $f(\mathbf{x}, \mathbf{y}) := G(\boldsymbol{\theta}, \mathbf{p}, \mathbf{s})$, $\mathbf{y}^*(\mathbf{x}) := \arg\max_{\mathbf{y}} f(\mathbf{x}, \mathbf{y})$ and $g(\mathbf{x}) := f(\mathbf{x}, \mathbf{y}^*(\mathbf{x}))$.

#### A.1.1 PROOF OF LEMMA 1

First we introduce some lemmas from previous literature.

**Lemma 2.** *(Lemma 1.2.3, Theorem 2.1.8 and Theorem 2.1.10 in Nesterov (2013))*

- *Suppose a function $h$ is $L_h$-gradient-Lipschitz and has a unique maximizer $\mathbf{x}^*$. Then, for any $\mathbf{x}$, we have:*

$$\frac{1}{2L_h}\|\nabla h(\mathbf{x})\|_2^2 \le h(\mathbf{x}^*) - h(\mathbf{x}) \le \frac{L_h}{2}\|\mathbf{x} - \mathbf{x}^*\|_2^2. \tag{15}$$

- *Suppose a function $h$ is $\tau_h$-strongly concave and has a unique maximizer $\mathbf{x}^*$. Then, for any $\mathbf{x}$, we have:*

$$\frac{\tau_h}{2}\|\mathbf{x} - \mathbf{x}^*\|_2^2 \le h(\mathbf{x}^*) - h(\mathbf{x}) \le \frac{1}{2\tau_h}\|\nabla h(\mathbf{x})\|_2^2. \tag{16}$$

From lemma 2 and the definition of $\epsilon$-stationary point (in definition 2) we can get the following lemma.

**Lemma 3.** *Suppose assumption 1 holds and $(\mathbf{x}_\epsilon, \mathbf{y}_\epsilon)$ is an $\epsilon$-stationary point of $\min_{\mathbf{x}} \max_{\mathbf{y}} f(\mathbf{x}, \mathbf{y})$, let $(\boldsymbol{\theta}_\epsilon, \mathbf{p}_\epsilon) = \mathbf{x}_\epsilon$ we have*

$$F(\boldsymbol{\theta}_\epsilon, \mathbf{s}_\epsilon) - \min_{\mathbf{s}} F(\boldsymbol{\theta}_\epsilon, \mathbf{s}) \le O(\frac{\epsilon^2}{\lambda^2}).$$

*Proof.*

$$F(\boldsymbol{\theta}_\epsilon, \mathbf{s}_\epsilon) - \min_{\mathbf{s}} F(\boldsymbol{\theta}_\epsilon, \mathbf{s}) \le \frac{1}{\tau}\|\nabla_{\mathbf{s}} F(\boldsymbol{\theta}_\epsilon, \mathbf{s}_\epsilon)\|^2 = \frac{1}{\lambda^2 \tau}\|\nabla_{\mathbf{y}} f(\mathbf{x}_\epsilon, \mathbf{y}_\epsilon)\|^2 \le O(\frac{\epsilon^2}{\lambda^2}),$$

here the first inequality is from Lemma 2 applied to $-F$ and the second inequality from definition 2. □

The following is a rephrase of theorem 2 in Lu & Mei (2024).

*Proof.* (proof of lemma 1) By Lemma 3 and the value of $\lambda$ we have

$$F(\boldsymbol{\theta}_\epsilon, \mathbf{s}_\epsilon) - \min_{\mathbf{s}} F(\boldsymbol{\theta}_\epsilon, \mathbf{s}) \le O(\epsilon^4).$$

Therefore, by Theorem 2 in Lu & Mei (2024) we have $\mathbb{E}[\|\nabla F(\boldsymbol{\theta}, \mathbf{p}^*(\boldsymbol{\theta}))\|] \le O(\epsilon)$ and Lemma 1 is proven. □

#### A.1.2 PROOF OF THEOREM 1

Based on Lemma 1, it suffices to prove that the algorithm 2 outputs an $\epsilon$-stationary point of $\min_{\mathbf{x}} \max_{\mathbf{y}} f(\mathbf{x}, \mathbf{y})$. In this section we will prove this conclusion.

First we introduce the smoothed function of $f$, which will be useful in the proof.

**Lemma 4.** *(Lemma C.2 in Zhang et al. (2024a)) Let $\mathbf{u}$ be uniformly sampled from the Euclidean sphere $\sqrt{d}\mathbb{s}^{d-1}$ and $\mathbf{v}$ be uniformly sampled from the Euclidean ball $\sqrt{d}\mathbb{B}^d = \{\mathbf{x} \in \mathbb{R}^d \mid \|\mathbf{x}\| \le \sqrt{d}\}$. For any function $f(\mathbf{x}) : \mathbb{R}^d \to \mathbb{R}$ and $\alpha > 0$, we define its zeroth-order gradient estimator as:*

$$\hat{\nabla} f_\alpha(\mathbf{x}) = \frac{f(\mathbf{x} + \alpha\mathbf{u}) - f(\mathbf{x} - \alpha\mathbf{u})}{2\alpha}\mathbf{u},$$

*and the smoothed function as:*

$$f_\alpha(\mathbf{x}) = \mathbb{E}_{\mathbf{v}}[f(\mathbf{x} + \alpha\mathbf{v})].$$

*The following properties hold:*

(i) $f_\alpha(\mathbf{x})$ *is differentiable and* $\mathbb{E}_{\mathbf{u}}[\hat{\nabla} f_\alpha(\mathbf{x})] = \nabla f_\alpha(\mathbf{x})$.

(ii) *If* $f(\mathbf{x})$ *is* $\ell$-*smooth, then we have that:*

$$\|\nabla f(\mathbf{x}) - \nabla f_\alpha(\mathbf{x})\| \le \frac{\ell}{2}\alpha d^{3/2}.$$

If we use $f(\mathbf{x}, \mathbf{y}; \xi)$ to denote a forward evaluation with random samples $\xi$ and let batch size $B = |\xi|$, then $f(\mathbf{x}, \cdot; \xi)$ is a function from $\mathbb{R}^d$ to $\mathbb{R}$ and $\ell$-smooth. The above lemma can be used on $f(\mathbf{x}, \cdot)$ and $f(\mathbf{x}, \cdot; \xi)$. We can define its smoothed function $f_\alpha(\mathbf{x}, \cdot; \xi)$ and has the properties above.

**Lemma 5.** *If assumption 1 holds, for* $f_\alpha$ *defined in Lemma 4,* $\nabla_{\mathbf{x}} f_\alpha(\mathbf{x}, \mathbf{y})$ *is* $\ell$-*continuous on* $\mathbf{y}$, *i.e.*

$$\|\nabla_{\mathbf{x}} f_\alpha(\mathbf{x}, \mathbf{y}_1) - \nabla_{\mathbf{x}} f_\alpha(\mathbf{x}, \mathbf{y}_2)\| \le \ell\|\mathbf{y}_1 - \mathbf{y}_2\|,$$

*for any* $\mathbf{x} \in \mathbb{R}^d, \mathbf{y}_1, \mathbf{y}_2 \in \mathbb{R}^{d'}$.

*Proof.*

$$\begin{aligned}
&\|\nabla_{\mathbf{x}} f_\alpha(\mathbf{x}, \mathbf{y}_1) - \nabla_{\mathbf{x}} f_\alpha(\mathbf{x}, \mathbf{y}_2)\|\\
=&\|\mathbb{E}_{\mathbf{v}}[f(\mathbf{x} + \alpha\mathbf{v}, \mathbf{y}_1)] - \mathbb{E}_{\mathbf{v}}[f(\mathbf{x} + \alpha\mathbf{v}, \mathbf{y}_2)]\|\\
\le&\mathbb{E}_{\mathbf{v}}\|f(\mathbf{x} + \alpha\mathbf{v}, \mathbf{y}_1) - f(\mathbf{x} + \alpha\mathbf{v}, \mathbf{y}_2)\|\\
\le&\ell\|\mathbf{y}_1 - \mathbf{y}_2\|.
\end{aligned}$$

Here the first inequality is from the convexity of norm and the second inequality is from the $\ell$-smoothness of $f$. $\qquad\square$

We first give the iteration complexity of the inner loop of Algorithm 2. Using the simplified notations we can write the update step in the inner loop as $\mathbf{y}_{t+1}^k = \mathbf{y}_t^k + \eta\nabla_{\mathbf{y}} f(\mathbf{x}^k, \mathbf{y}_t^k; \xi_t)$. We use $B_1, B_2$ to denote the batch size for the inner loop and outer loop, respectively. But finally we will prove that they are in fact of the same order.

**Lemma 6.** *In Algorithm 2, by setting* $\eta = 1/2\ell$, $T = O(\kappa \log(\frac{1}{\epsilon}))$ *and* $B_1 = O(\epsilon^{-2})$ *we have*

$$\mathbb{E}[\|\mathbf{y}_T^k - \mathbf{y}^*(\mathbf{x}^k)\|^2] \le \epsilon^2$$

*in outer loop* $k$.

*Proof.*

$$\begin{aligned}
&\|\mathbf{y}_{t+1}^k - \mathbf{y}^*(\mathbf{x}^k)\|^2\\
=&\|\mathbf{y}_t^k + \eta\nabla_{\mathbf{y}} f(\mathbf{x}^k, \mathbf{y}_t^k; \xi_t) - \mathbf{y}^*(\mathbf{x}^k)\|^2\\
=&\|\mathbf{y}_t^k - \mathbf{y}^*(\mathbf{x}^k)\|^2 + 2\eta\langle\nabla_{\mathbf{y}} f(\mathbf{x}^k, \mathbf{y}_t^k; \xi_t), \mathbf{y}_t^k - \mathbf{y}^*(\mathbf{x}^k)\rangle + \eta^2\|\nabla_{\mathbf{y}} f(\mathbf{x}^k, \mathbf{y}_t^k; \xi_t)\|^2.
\end{aligned}$$

Now taking expectations on both sides we have

$$\mathbb{E}[\|\mathbf{y}_{t+1}^k - \mathbf{y}^*(\mathbf{x}^k)\|^2]$$

$$\le\mathbb{E}[\|\mathbf{y}_t^k - \mathbf{y}^*(\mathbf{x}^k)\|^2] + 2\eta\mathbb{E}[\langle\nabla_{\mathbf{y}} f(\mathbf{x}^k, \mathbf{y}_t^k), \mathbf{y}_t^k - \mathbf{y}^*(\mathbf{x}^k)\rangle] + \eta^2(\mathbb{E}[\|\nabla_{\mathbf{y}} f(\mathbf{x}^k, \mathbf{y}_t^k)\|^2] + \frac{\sigma^2}{B_1})$$

$$\le\mathbb{E}[\|\mathbf{y}_t^k - \mathbf{y}^*(\mathbf{x}^k)\|^2] - 2\eta\mathbb{E}[f(\mathbf{x}^k, \mathbf{y}^*(\mathbf{x}^k)) - f(\mathbf{x}^k, \mathbf{y}_t^k)] + 2\ell\eta^2\mathbb{E}[f(\mathbf{x}^k, \mathbf{y}^*(\mathbf{x}^k)) - f(\mathbf{x}^k, \mathbf{y}_t^k)] + \frac{\eta^2\sigma^2}{B_1}$$

$$=\mathbb{E}[\|\mathbf{y}_t^k - \mathbf{y}^*(\mathbf{x}^k)\|^2] - \frac{1}{2\ell}\mathbb{E}[f(\mathbf{x}^k, \mathbf{y}^*(\mathbf{x}^k)) - f(\mathbf{x}^k, \mathbf{y}_t^k)] + \frac{\sigma^2}{4\ell^2 B_1}$$

$$\le\mathbb{E}[\|\mathbf{y}_t^k - \mathbf{y}^*(\mathbf{x}^k)\|^2] - \frac{\tau}{4\ell}\mathbb{E}[\|\mathbf{y}_t^k - \mathbf{y}^*(\mathbf{x}^k)\|^2] + \frac{\sigma^2}{4\ell^2 B_1}.$$

The first inequality is from Assumption 1, second and last inequalities from Lemma 2 and the equation is from the value of $\eta$.

In order for $\mathbb{E}[\|\mathbf{y}_T^k - \mathbf{y}^*(\mathbf{x}^k)\|^2] \leq \epsilon^2$ we need $T = O(\kappa \log(\frac{1}{\epsilon}))$ and $B_1 = O(\epsilon^{-2})$. $\qquad\square$

The following lemma is from Theorem 1 in Malladi et al. (2023).

**Lemma 7.** *If Assumption 2 holds, there exists a constant $\gamma = \boldsymbol{\theta}(r)$ such that*

$$\mathbb{E}[\hat{\nabla}_{\mathbf{x}}f(\mathbf{x}^k,\mathbf{y}^{k+1};\xi)^T H(\mathbf{x}^k,\mathbf{y}^{k+1})\hat{\nabla}_{\mathbf{x}}f(\mathbf{x}^k,\mathbf{y}^{k+1};\xi)] \leq \ell\gamma\mathbb{E}[\|\nabla_{\mathbf{x}}f(\mathbf{x}^k,\mathbf{y}^{k+1};\xi)\|^2].$$

Finally, we give the proof for Theorem 1. In this part we assume both $\boldsymbol{\theta}$ and $\mathbf{p}$ updates with zeroth order gradient for the convenience of analysis and this does not change the order of the total complexity.

*Proof.* (**proof of Theorem 1**)

From Assumption 2, taking expectation conditioning on $\mathbf{x}^k$ and $\mathbf{y}^{k+1}$ we have

$$\begin{aligned}
\mathbb{E}[g(\mathbf{x}^{k+1})] \leq &g(\mathbf{x}^k) - \zeta\langle\nabla_{\mathbf{x}}g(\mathbf{x}^k), \mathbb{E}[\hat{\nabla}_{\mathbf{x}}f(\mathbf{x}^k,\mathbf{y}^{k+1};\xi)]\rangle \\
&+ \frac{\zeta^2}{2}\mathbb{E}[\hat{\nabla}_{\mathbf{x}}f(\mathbf{x}^k,\mathbf{y}^{k+1};\xi)^T H(\mathbf{x}^k,\mathbf{y}^{k+1})\hat{\nabla}_{\mathbf{x}}f(\mathbf{x}^k,\mathbf{y}^{k+1};\xi)] \\
\leq &g(\mathbf{x}^k) - \zeta\langle\nabla_{\mathbf{x}}g(\mathbf{x}^k), \nabla_x f_\alpha(\mathbf{x}^k,\mathbf{y}^{k+1})\rangle + \frac{\zeta^2}{2}\ell\gamma\mathbb{E}[\|\nabla_{\mathbf{x}}f(\mathbf{x}^k,\mathbf{y}^{k+1};\xi)\|^2]
\end{aligned}$$

Let us bound the inner product term:

$$\begin{aligned}
&-\zeta\langle\nabla_{\mathbf{x}}g(\mathbf{x}^k), \nabla_x f_\alpha(\mathbf{x}^k,\mathbf{y}^{k+1})\rangle \\
\leq &-\zeta\langle\nabla_{\mathbf{x}}f(\mathbf{x}^k,\mathbf{y}^*(\mathbf{x}^k)) - \nabla_{\mathbf{x}}f_\alpha(\mathbf{x}^k,\mathbf{y}^*(\mathbf{x}^k)) + \nabla_{\mathbf{x}}f_\alpha(\mathbf{x}^k,\mathbf{y}^*(\mathbf{x}^k)) \\
&-\nabla_x f_\alpha(\mathbf{x}^k,\mathbf{y}^{k+1}) + \nabla_x f_\alpha(\mathbf{x}^k,\mathbf{y}^{k+1}), \nabla_x f_\alpha(\mathbf{x}^k,\mathbf{y}^{k+1})\rangle \\
\leq &\frac{1}{\ell\gamma}\|\nabla_{\mathbf{x}}f(\mathbf{x}^k,\mathbf{y}^*(\mathbf{x}^k)) - \nabla_{\mathbf{x}}f_\alpha(\mathbf{x}^k,\mathbf{y}^*(\mathbf{x}^k))\|^2 + \frac{\zeta^2\ell\gamma}{4}\|\nabla_x f_\alpha(\mathbf{x}^k,\mathbf{y}^{k+1})\|^2 \\
&+\frac{1}{\ell\gamma}\|\nabla_{\mathbf{x}}f_\alpha(\mathbf{x}^k,\mathbf{y}^*(\mathbf{x}^k)) - \nabla_x f_\alpha(\mathbf{x}^k,\mathbf{y}^{k+1})\|^2 + \frac{\zeta^2\ell\gamma}{4}\|\nabla_x f_\alpha(\mathbf{x}^k,\mathbf{y}^{k+1})\|^2 \\
&-\zeta\langle\nabla_x f_\alpha(\mathbf{x}^k,\mathbf{y}^{k+1}), \nabla_x f_\alpha(\mathbf{x}^k,\mathbf{y}^{k+1})\rangle \\
\leq &\frac{\alpha^2\ell^2 d^3}{4\ell\gamma} + \frac{\ell^2}{\ell\gamma}\|\mathbf{y}^*(\mathbf{x}^k) - \mathbf{y}^{k+1}\|^2 + \frac{\zeta^2\ell\gamma}{2}\|\nabla_x f_\alpha(\mathbf{x}^k,\mathbf{y}^{k+1})\|^2 \\
&-\zeta\langle\nabla_x f_\alpha(\mathbf{x}^k,\mathbf{y}^{k+1}), \nabla_x f_\alpha(\mathbf{x}^k,\mathbf{y}^{k+1})\rangle.
\end{aligned}$$

Here the last inequality is from Lemma 4 and Lemma 5.

Now back to the original inequality, taking expectations over all the randomness in the algorithm we have

$$\begin{aligned}
&\zeta(1 - \frac{\zeta\ell\gamma}{2})\mathbb{E}[\|\nabla_x f_\alpha(\mathbf{x}^k,\mathbf{y}^{k+1})\|^2] \\
\leq &\mathbb{E}[g(\mathbf{x}^k) - g(\mathbf{x}^{k+1})] + \frac{\ell}{\gamma}\mathbb{E}[\|\mathbf{y}^*(\mathbf{x}^k) - \mathbf{y}^{k+1}\|^2] + \frac{\zeta^2\ell\gamma}{2}\mathbb{E}[\|\nabla_{\mathbf{x}}f(\mathbf{x}^k,\mathbf{y}^{k+1};\xi)\|^2] + \frac{\alpha^2\ell d^3}{4\gamma} \\
\leq &\mathbb{E}[g(\mathbf{x}^k) - g(\mathbf{x}^{k+1})] + \frac{\ell}{\gamma}\mathbb{E}[\|\mathbf{y}^*(\mathbf{x}^k) - \mathbf{y}^{k+1}\|^2] + \frac{\zeta^2\ell\gamma}{2}\mathbb{E}[\|\nabla_{\mathbf{x}}f(\mathbf{x}^k,\mathbf{y}^{k+1})\|^2] + \frac{\zeta^2\ell\gamma\sigma^2}{2B_2} + \frac{\alpha^2\ell d^3}{4\gamma},
\end{aligned}$$

where the last inequality is from Assumption 1.

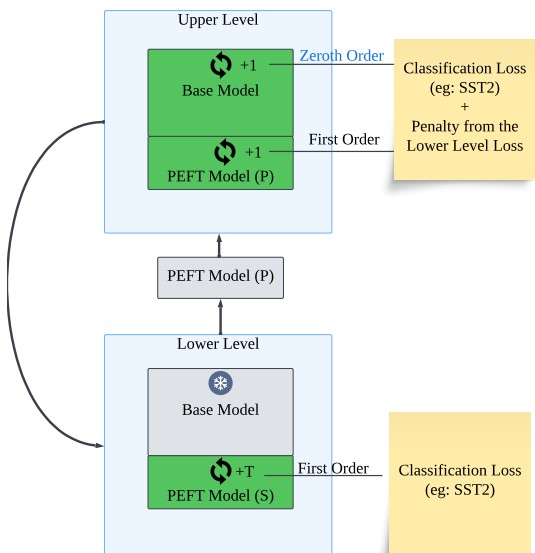

Figure 3: Pipeline for Zeroth-order-first-order bilevel method.

On the other hand, from Lemma 4, by letting $\zeta = \frac{1}{2\ell\gamma}$ we have

$$\mathbb{E}[\|\nabla_x f(\mathbf{x}^k, \mathbf{y}^{k+1})\|^2]$$

$$\leq 2\mathbb{E}[\|\nabla_x f_\alpha(\mathbf{x}^k, \mathbf{y}^{k+1})\|^2] + \frac{\alpha^2\ell^2(d+d')^3}{2}$$

$$\leq \frac{16}{3}\ell\gamma\mathbb{E}[g(\mathbf{x}^k) - g(\mathbf{x}^{k+1})] + \frac{16}{3}\ell^2\mathbb{E}[\|\mathbf{y}^*(\mathbf{x}^k) - \mathbf{y}^{k+1}\|^2]$$

$$+ \frac{2}{3}\mathbb{E}[\|\nabla_{\mathbf{x}} f(\mathbf{x}^k, \mathbf{y}^{k+1})\|^2] + \frac{2\sigma^2}{3B_2} + \frac{11}{6}\alpha^2\ell^2(d+d')^3$$

$$\Rightarrow \mathbb{E}[\|\nabla_x f(\mathbf{x}^k, \mathbf{y}^{k+1})\|^2] \leq 16\ell\gamma\mathbb{E}[g(\mathbf{x}^k) - g(\mathbf{x}^{k+1})] + 16\ell^2\mathbb{E}[\|\mathbf{y}^*(\mathbf{x}^k) - \mathbf{y}^{k+1}\|^2]$$

$$+ \frac{2\sigma^2}{B_2} + \frac{11}{2}\alpha^2\ell^2(d+d')^3.$$

Taking summation of $k$ from 1 to $K$ we have

$$\frac{1}{K}\sum_{k=1}^{K+1}\mathbb{E}[\|\nabla_x f(\mathbf{x}^k, \mathbf{y}^{k+1})\|^2]$$

$$\leq \frac{16\ell\gamma}{K}\mathbb{E}[g(\mathbf{x}^1) - g(\mathbf{x}^{K+1})] + \frac{16\ell^2}{K}\sum_{k=1}^{K}\mathbb{E}[\|\mathbf{y}^*(\mathbf{x}^k) - \mathbf{y}^{k+1}\|^2] + \frac{2\sigma^2}{B_2} + \frac{11}{2}\alpha^2\ell^2(d+d')^3$$

$$\leq \frac{16\ell\gamma}{K}\mathbb{E}[g(\mathbf{x}^1) - \min_{\mathbf{x}} g(\mathbf{x})] + \frac{16\ell^2}{K}\sum_{k=1}^{K}\mathbb{E}[\|\mathbf{y}^*(\mathbf{x}^k) - \mathbf{y}^{k+1}\|^2] + \frac{2\sigma^2}{B_2} + \frac{11}{2}\alpha^2\ell^2(d+d')^3.$$

Thus, by setting parameters as in Theorem 1 we have $\min_k \mathbb{E}[\|\nabla_x f(\mathbf{x}^k, \mathbf{y}^{k+1})\|^2] \leq \epsilon^2$.

On the other hand, since

$$\mathbb{E}[\|\nabla_x f(\mathbf{x}^k, \mathbf{y}^{k+1})\|^2] = \mathbb{E}[\|\nabla_x f(\mathbf{x}^k, \mathbf{y}^{k+1}) - \nabla_{\mathbf{y}} f(\mathbf{x}^k, \mathbf{y}^*(\mathbf{x}^k))\|^2] \leq \ell^2\mathbb{E}[\|\mathbf{y}^{k+1} - \mathbf{y}^*(\mathbf{x}^k)\|^2],$$

similar to Lemma 6 we have $\mathbb{E}[\|\nabla_x f(\mathbf{x}^k, \mathbf{y}^{k+1})\|^2] \leq \epsilon^2$ by setting $T = O(\kappa \log(\frac{\kappa}{\epsilon}))$ and $B_1 = O(\epsilon^{-2})$. $\qquad\square$

## B    EXPERIMENTAL SETUP

To recall the proposed Algorithm 2, we present a pipeline of the proposed Algorithm 2 in figure 3.

## B.1 Single-Task experiments

Following MeZO (Malladi et al., 2023), we evaluate our approach on a range of classification and multiple-choice tasks. In this setting, training and testing are conducted on the same task.

### B.1.1 Tasks

We use the following tasks for evaluating the fine-tuning capabilities of Bilevel-ZOFO in a single-task setting.

**BoolQ (Clark et al., 2019):** A yes/no question-answering task where each question is paired with a paragraph that contains the answer.

**CB (Wang et al., 2019):** The CommitmentBank task involves determining whether a given sentence in context entails, contradicts, or is neutral to a premise.

**COPA (Roemmele et al., 2011):** The Choice of Plausible Alternatives (COPA) task requires selecting the most plausible cause or effect from two alternatives for a given premise.

**ReCoRD: (Zhang et al., 2018)** The Reading Comprehension with Commonsense Reasoning Dataset (ReCoRD) is a cloze-style task where models must predict masked-out entities in text based on the surrounding context.

**RTE (Wang, 2018):** The Recognizing Textual Entailment (RTE) task involves determining whether a given hypothesis is entailed by a provided premise.

**SST2 (Wang, 2018):** The Stanford Sentiment Treebank (SST-2) task focuses on binary sentiment classification of sentences as positive or negative.

**WiC (Pilehvar & Camacho-Collados, 2018):** The Word-in-Context (WiC) task involves determining whether the same word is used in the same sense in two different sentences.

**WinoGrande (Sakaguchi et al., 2021):** A commonsense reasoning task where the goal is to resolve pronoun references in ambiguous sentences by identifying the correct antecedent.

**WSC (Levesque et al., 2012):** The Winograd Schema Challenge (WSC) tests a model's ability to resolve pronoun references in sentences, requiring commonsense reasoning.

**SQuAD (Rajpurkar, 2016):** The Stanford Question Answering Dataset (SQuAD) is a reading comprehension task where models must answer questions based on a given passage of text.

### B.1.2 PEFT Variants

We utilize three PEFT techniques—prompt-tuning (Lester et al., 2021), prefix-tuning (Li & Liang, 2021), and LoRA (Hu et al., 2022)—for lower-level training to evaluate bilevel-ZOFO across various conditions and resource constraints.

1. **LoRA:** For all single-task LoRA experiments, we set $r = 8$ and $\alpha = 16$.
2. **Prefix Tuning:** We use 5 prefix tokens across all experiments.
3. **Prompt Tuning:** We configure 10 soft prompt tokens for every experiment.

### B.1.3 Hyperparameter Search

Given resource limitations, we focus on sweeping only the learning rate as the key hyperparameter. For MeZO and first-order PEFT experiments, we explore learning rates from the set $\{1e-2, 1e-3, 1e-4, 1e-5, 1e-6\}$. For Bilevel-ZOFO, we sweep both the upper-level and lower-level learning rates: $\text{lr}_{\text{upper}} \in \{1e-4, 1e-5, 1e-6\}$ and $\text{lr}_{\text{lower}} \in \{1e-2, 1e-3, 1e-4, 1e-5\}$. We perform all

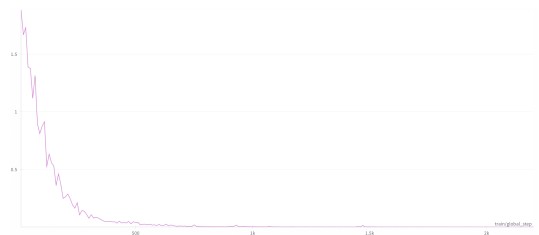

Figure 4: Training loss for the lower-level objective of the bilevel framework with Lora as the PEFT model.

experiments in tables 4 and 5 using three random seeds and report the average and standard deviation. We also set $\epsilon = 1e - 3$, following MeZO Malladi et al. (2023).

### B.1.4 TRAINING

All experiments used a batch size of 8 and were conducted in bfloat16 precision on a single A6000 Ada 48GB GPU. MeZO was run for 10,000 steps, while FO and Bilevel-ZOFO methods were run for 5,000 steps. Our implementation builds upon MeZO's codebase, and memory profiling as well as latency calculations are based on their framework.

For each task, 1000 examples are randomly sampled for training, 500 for validation, and 1000 for testing. For bilevel-ZOFO, the training set is split into upper-level and lower-level subsets with a 1:2 ratio. During each lower-level update, only the PEFT parameters are optimized, while in the upper-level step, the entire model is fine-tuned using zeroth-order gradient approximation. We set $\lambda = 10000$ and perform 20 lower-level updates between each upper-level update for all bilevel-ZOFO experiments.

All experiments use the Adam optimizer (Kingma & Ba, 2015),including baselines and both lower-level and upper-level optimizers. No weight decay was applied, and the models were trained with a constant learning rate schedule. Batch size is set to 16 for all experiments. We load all models in bfloat16. We find the best performing model based on validation loss and report test results from that checkpoint. We report the test accuracy or F1-score based on the test dataset being imbalanced or not.

We fix the memory budget of each step across bilevel-ZOFO and the baselines. We train zeroth-order methods for 10,000 steps, and bilevel-ZOFO and first-order methods for 5000 steps. We use A6000ada 48GPUs in our experiments. We load all models in bfloat16.

### B.2 RESULTS

Figure 4 presents the training loss for the lower-level objective of the bilevel framework with Lora as the PEFT model. As shown, consistent with the guarantees provided by our theoretical analysis, Bilevel-ZOFO converges.

Table 4 presents the test metrics when applying bilevel-ZOFO and baselines to fine-tune OPT-1.3B (Zhang et al., 2022) on a downstream task.

| Trainer | Mode | BoolQ | CB | Copa | ReCoRD | RTE | SST2 | WIC | WinoGrande | WSC | Average |
|---|---|---|---|---|---|---|---|---|---|---|---|
| MeZO | ft | 0.6927 ± 0.0660 | 0.7767 ± 0.1162 | 0.7000 ± 0.0289 | 0.6980 ± 0.0053 | 0.6587 ± 0.0271 | 0.8214 ± 0.0042 | 0.5543 ± 0.0146 | 0.5480 ± 0.0108 | 0.5054 ± 0.0056 | 0.6617 ± 0.0321 |
| | lora | 0.6860 ± 0.0012 | 0.7607 ± 0.0515 | 0.7200 ± 0.0058 | 0.7083 ± 0.0049 | 0.6755 ± 0.0110 | 0.8501 ± 0.0067 | 0.5549 ± 0.0057 | 0.5607 ± 0.0050 | 0.5570 ± 0.0000 | 0.6748 ± 0.0102 |
| | prefix | 0.6573 ± 0.0379 | 0.7945 ± 0.0309 | 0.7033 ± 0.0208 | 0.7047 ± 0.0010 | 0.6972 ± 0.0055 | 0.8218 ± 0.0127 | 0.5622 ± 0.0127 | 0.5370 ± 0.0137 | 0.5105 ± 0.1313 | 0.6654 ± 0.0285 |
| | prompt | 0.6260 ± 0.0056 | 0.5821 ± 0.0179 | 0.7067 ± 0.0058 | 0.7070 ± 0.0053 | 0.5415 ± 0.0063 | 0.7463 ± 0.0218 | 0.5574 ± 0.0048 | 0.5556 ± 0.0038 | 0.4654 ± 0.0618 | 0.6098 ± 0.0159 |
| | average | 0.6655 | 0.7285 | 0.7075 | 0.7045 | 0.6432 | 0.8099 | 0.5572 | 0.5503 | 0.5096 | 0.6529 ± 0.0217 |
| FO | lora | 0.7403 ± 0.0055 | 0.8512 ± 0.0412 | 0.7500 ± 0.0058 | 0.7206 ± 0.0035 | 0.7292 ± 0.0165 | 0.9258 ± 0.0032 | 0.6463 ± 0.0276 | 0.5806 ± 0.0055 | 0.6474 ± 0.0200 | 0.7324 ± 0.0143 |
| | prefix | 0.7300 ± 0.0035 | 0.8571 ± 0.0644 | 0.7167 ± 0.0115 | 0.7093 ± 0.0032 | 0.7136 ± 0.0110 | 0.8133 ± 0.0050 | 0.5387 ± 0.0050 | 0.5980 ± 0.0029 | 0.5705 ± 0.0294 | 0.6941 ± 0.0141 |
| | prompt | 0.7150 ± 0.0156 | 0.7142 ± 0.0714 | 0.7466 ± 0.0115 | 0.7163 ± 0.0063 | 0.6936 ± 0.0185 | 0.8016 ± 0.0779 | 0.5386 ± 0.0197 | 0.5980 ± 0.0090 | 0.5062 ± 0.0434 | 0.6700 ± 0.0306 |
| | average | 0.7284 | 0.8075 | 0.7378 | 0.7154 | 0.7121 | 0.8470 | 0.5745 | 0.5922 | 0.5747 | 0.6982 ± 0.0197 |
| Ours | lora | 0.7433 ± 0.0191 | 0.9167 ± 0.0103 | 0.7400 ± 0.0200 | 0.7183 ± 0.0031 | 0.7401 ± 0.0108 | 0.9331 ± 0.0020 | 0.6447 ± 0.0218 | 0.5903 ± 0.0058 | 0.6428 ± 0.0855 | 0.7410 ± 0.0209 |
| | prefix | 0.7340 ± 0.0095 | 0.8690 ± 0.0206 | 0.7267 ± 0.0153 | 0.7140 ± 0.0044 | 0.7304 ± 0.0091 | 0.8550 ± 0.0178 | 0.6317 ± 0.0282 | 0.5710 ± 0.0130 | 0.5810 ± 0.0338 | 0.7125 ± 0.0179 |
| | prompt | 0.7367 ± 0.0850 | 0.7679 ± 0.0644 | 0.7633 ± 0.0058 | 0.7257 ± 0.0153 | 0.6867 ± 0.0208 | 0.8335 ± 0.0779 | 0.6267 ± 0.0462 | 0.5900 ± 0.0173 | 0.5133 ± 0.1493 | 0.6938 ± 0.0536 |
| | average | 0.7380 | 0.8512 | 0.7433 | 0.7193 | 0.7191 | 0.8739 | 0.6344 | 0.5838 | 0.5790 | 0.7158 ± 0.0308 |

Table 4: Single-Task Experiments on OPT-1.3B with 1000 samples. Values correspond to mean across three random seeds. FO: First-Order. FT: full-model fine-tuning.

| Trainer | Mode | BoolQ | ReCoRD | SQuAD | SST2 | Average |
|---------|------|-------|--------|-------|------|---------|
| MeZO | ft | 0.7915 ± 0.0516 | 0.7890 ± 0.0001 | 0.7737 ± 0.1634 | 0.8646 ± 0.0216 | 0.8047 |
|  | lora | 0.8020 ± 0.0014 | 0.7970 ± 0.0001 | 0.7412 ± 0.0013 | 0.8529 ± 0.0117 | 0.7983 |
|  | prefix | 0.7830 ± 0.0131 | 0.7905 ± 0.0007 | 0.7093 ± 0.0207 | 0.8364 ± 0.0010 | 0.7798 |
|  | prompt | 0.7787 ± 0.0049 | 0.7935 ± 0.0007 | 0.7014 ± 0.0451 | 0.8246 ± 0.0216 | 0.7746 |
| FO | lora | 0.8420 ± 0.0104 | 0.7920 ± 0.0053 | 0.8197 ± 0.0043 | 0.9557 ± 0.0007 | 0.8524 |
|  | prefix | 0.7783 ± 0.0021 | 0.8013 ± 0.0012 | 0.7946 ± 0.0419 | 0.9243 ± 0.0053 | 0.8246 |
|  | prompt | 0.8083 ± 0.0142 | 0.8023 ± 0.0074 | 0.7805 ± 0.0633 | 0.9284 ± 0.0072 | 0.8299 |
| Ours | lora | 0.8473 ± 0.0025 | 0.8290 ± 0.0044 | 0.8160 ± 0.0041 | 0.9629 ± 0.0053 | 0.8638 |
|  | prefix | 0.8193 ± 0.0127 | 0.8067 ± 0.0065 | 0.8090 ± 0.0302 | 0.9382 ± 0.0064 | 0.8433 |
|  | prompt | 0.8145 ± 0.0012 | 0.8108 ± 0.0065 | 0.7960 ± 0.0028 | 0.9222 ± 0.0039 | 0.8359 |

Table 5: Single-Task Experiments on Llama2-7B with 1000 samples. Values correspond to mean and std across three random seeds. FO: First-Order. FT: full-model fine-tuning

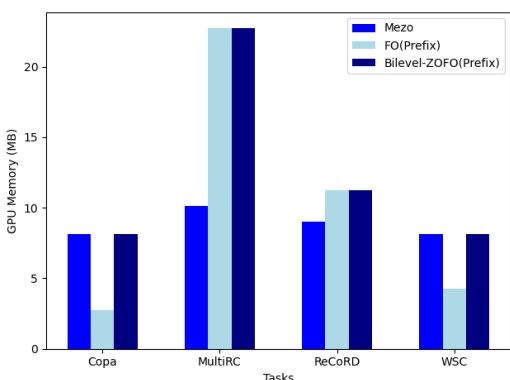

Figure 5: Memory consumption of MeZO and first-order PEFT methods varies across tasks, with one occasionally surpassing the other. Our Bilevel-ZOFO method demonstrates comparable memory usage to both baselines. Values correspond to memory usage for fine-tuning OPT1.3b Zhang et al. (2022) on each task using a batch size of 8 and on a singel A6000ada 48GB GPU.

Table 5 demonstrates the results for fine-tuning Llama2-7b (Touvron et al., 2023) on various classification and open-ended generation tasks.

### B.3 MEMORY PROFILING AND WALL CLOCK TIME ANALYSIS

Figure 5 demonstrates the memory profiling of Bilevel-ZOFO, MeZO and First-order prefix tuning on four different tasks. Memory consumption of MeZO and first-order PEFT methods varies across tasks, with one occasionally surpassing the other. Each lower-level update in our method matches that of the corresponding PEFT method. Similarly, each upper-level update requires the greater memory usage between MeZO and PEFT under comparable settings. As a result, the total memory requirement of our method corresponds to the maximum memory usage of the PEFT and MeZO experiments. Nonetheless, as demonstrated in Table 4, our method outperforms both PEFT and MeZO on average.

We also present a wall-clock time analysis of bilevel-ZOFO compared to the baseline. As shown in Table 6, similar to MeZO Malladi et al. (2023), we observe that zeroth-order steps exhibit higher latency compared to first-order steps. The results indicate that our bilevel-ZOFO achieves comparable delays to the FO-PEFT method while significantly reducing step duration compared to MeZO. Moreover, as highlighted in Table 1, bilevel-ZOFO outperforms both methods on average.

### B.4 MULTI-TASK EXPERIMENTS

In this section we explain the experimental details of mutil-task experiments.

Table 6: Wallclock time per step of different training methods when finetuning OPT1.3b. The values are measured on a single A6000ada 48GB GPU. The wallclock time is averaged over 3 different runs that produced the values of Table 1. We use a batch size of 8 for all experiments.

| Task | MeZO | FO Prefix-Tuning | Bilevel-ZOFO (Prefix) |
|------|------|------------------|------------------------|
| Copa | 0.299 | 0.127 | 0.135 |
| MultiRC | 0.622 | 0.474 | 0.502 |
| WSC | 0.278 | 0.120 | 0.164 |

### B.4.1 META-TASKS

Following the methodology of Min et al. (2022), we evaluate the performance of bilevel-ZOFO as a fast and efficient meta-learning algorithm. We perform experiments using four of the distinct meta-learning settings outlined in MetaICL (Min et al., 2022): classification-to-classification, non-classification-to-classification, QA-to-QA, and non-QA-to-QA. Each of these *meta-learning tasks* includes a set of training sub-tasks and a different set of test sub-tasks. The sub-tasks are sourced from CROSSFIT (Ye et al., 2021) and UNIFIEDQA (Khashabi et al., 2020), comprising a total of 142 unique sub-tasks. These sub-tasks cover a variety of problems, including text classification, question answering, and natural language understanding, all in English. Table 7 shows the number of tasks in each training and testing meta-learning setting and the total number of examples in each training task.

| Meta-train Setting | # tasks | # examples | Target Setting | # tasks |
|--------------------|---------|-----------|----------------|---------|
| Classification | 43 | 384,022 | Classification | 20 |
| Non-Classification | 37 | 368,768 | | |
| QA | 37 | 486,143 | QA | 22 |
| Non-QA | 33 | 521,342 | | |

Table 7: Details of four different meta-learning settings. Each row indicates meta-training/target tasks for each setting. There is no overlap between the training and test tasks.

See Tables 14 and 15 of MetaICL (Min et al., 2022) for a list of all sub-tasks.

### B.4.2 BASELINES

We use GPT2-Large Radford et al. (2019) as the base model for these experiments.We compare our method against several baseline approaches:

- **MetaICL** (Min et al., 2022): A method for meta-learning with in-context learning. MetaICL tunes all the parameters of the base model using the first-order method. In both training and testing, the model is given $k$ demonstration examples, $(a_1, b_1), \ldots, (a_k, b_k)$, where $b_i$ represents either classification labels or possible answers in question-answering tasks, along with one test example $(a, b)$. The input is formed by concatenating the demonstration examples $a_1, b_1, \ldots, a_k, b_k, a$. The model then computes the conditional probability of each label, and the label with the highest probability is selected as the prediction.

- **Zero-shot**: This method uses the pretrained language model (LM) without any tuning, performing zero-shot inference without any demonstration examples.

- **In-context Learning (ICL)**: This method uses the pretrained LM with in-context learning by conditioning on a concatenation of $k$ demonstration examples and 1 actual test sample similar to MetaICL.

We sample 768 examples from each training sub-task. We use these samples to train MetaICL in their original setting for 30,000 steps. This includes learning rate of $1e - 5$, batch size of $1$ on $8$ GPUs, 8-bit Adam optimizer and fp16 half precision. See MetaICL(Min et al., 2022) for full details. To train our method, we split the training dataset of each sub-task to two subsets, 256 samples as the development dataset for upper-level updates and 512 samples for lower-level training. For each outer iteration of our method, we randomly sample a subset of 5 training tasks. We perform 10 lower-level

updates between each pair of upper-level updates. To keep bilevel-ZOFO as lightweight as possible, unlike MetaICL, we do not include demonstration examples in the inputs. Since bilevel-ZOFO uses significantly less memory and has much faster updates compared to MetaICL, theoretically we are able to train it for many more iterations within the same total training duration as MetaICL. However, due to resource constraints, we only train bilevel-ZOFO for 50,000 iterations. Similar to Malladi et al. (2023), we did not observe a plateau in performance for bilevel-ZOFO, indicating that further training can yield additional improvements. We use Adam optimizer and a learning rate of $1e-6$ for both upper and lower-level training. We employ a batch size of $4$ and train on a single rtx6000ada GPU.

For both ICL and MetaICL, during the testing phase the model is given $k = 4$ demonstration examples for each test data point. We don't use demonstration examples in test samples for bilevel-ZOFO evaluation. We evaluate the zero-shot capabilities of our method as well as the performance of the final model LoRA-tuned for 10 additional iterations on $4$ demonstration samples from each class of each test sub-task. Similar to Min et al. (2022), we report **Macro-averaged F1** as the evaluation metric.

