# OpenReview forum: "Bilevel ZOFO: Bridging Parameter-Efficient and Zeroth-Order Techniques for Efficient LLM Fine-Tuning and Meta-Training"
_ICLR.cc/2025/Conference — Submitted to ICLR 2025_

### Official Review · Reviewer_mLL4 · 2024-11-03

**Soundness:** 3
**Presentation:** 3
**Contribution:** 3
**Rating:** 5
**Confidence:** 5

**Summary:**

The paper introduces a bilevel optimization method called Bilevel to efficiently finetune LLMs by combining PEFT and ZO optimization techniques. The motivation is to address the computational inefficiencies of traditional finetuning, which requires backpropagation and substantial memory. The primary research question centers on whether PEFT can enhance ZO fine-tuning for both single-task and multitask settings without requiring extensive computational resources. Bilevel ZOFO uses a two-level optimization framework that applies ZO for full model fine-tuning at the upper level and PEFT at the lower level to minimize computational cost while maintaining high performance. Experiments demonstrate that Bilevel ZOFO outperforms both individual ZO and PEFT methods on single tasks and achieves competitive results in multitask learning with reduced computational demand.

**Strengths:**

The paper has several major strengths:

1. Bilevel ZOFO is effectively extended to multitask learning scenarios, allowing models to handle multiple tasks simultaneously. This is beneficial in resource-limited environments where large, labeled datasets are scarce, such as in medical domains.

2. The paper provides theoretical guarantees for the convergence of the Bilevel ZOFO method.

3. Empirical results demonstrate that Bilevel ZOFO consistently outperforms both standalone ZO and PEFT approaches in single-task finetuning, and it is competitive with SOTA meta-learning methods for multitask scenarios.

**Weaknesses:**

The weaknesses of this paper are as follows:

1. The mathematical notations can be further improved to follow the convention. For example, when denoting a vector, usually bold font is preferred, such $\boldsymbol{\theta} \in \mathbb{R}^d$ instead of $\theta \in \mathbb{R}^d$. This helps the reader differentiate vectors from scalars and improves readability.

2. Typo at the end of line 269 "Sso" should be 'So'

3. In Eq. 2, I am not fully convinced why we need to train the PEFT module p and the base model parameter $\theta$ at the same time? Wouldn't avoiding training $\theta$ be the motivation for using PEFT modules?

4. The motivation for using ZO method in Algorithm 2 is not very clear to me. You may notice that as long as you need to do backpropagation, even if it is parameter efficient finetuning, the computational graph of all the network (from shallow to deep layers) should all be stored and thus, the memory consumption is still very high. Wouldn't this contradict the motivation for using ZO method? Why wouldn't the authors use ZO method throughout the algorithm, namely, use ZO method to calculate the gradient of the PEFT modules as well.

5. As a follow-up question of 4, this paper does not report the efficiency comparison of any method. Specifically, the memory consumption, time consumption, and the convergence speed should be carefully measured and reported. I am afraid that the combination of ZO and PEFT in a way presented in this paper would harm the efficiency of both methods.

**Questions:**

Please see my questions in the weakness column.

---

> ### Author Response · Authors · 2024-11-21
> **Response to Reviewer mLL4**
>
> We thank the reviewer for their detailed review of our method. We will address the concerns raise below:
>
> ## 1- Notations and typo
> Thank you for the comment. We have bolded the vectors in the revised version, marked in blue for easy reference. Please refer to the updated version. We have fixed the typo you mentioned in the revised version.
>
> ## 2- Why Tune both the PEFT and base model?
>
> While PEFT methods effectively reduce training costs and memory usage, they do not always achieve the same level of task-specific performance as full model fine-tuning (as noted in studies such as Hu et al., 2022; Li & Liang, 2021; Zaken et al., 2022). Therefore, in our approach, we aim to improve performance by tuning the full model in addition to the PEFT module.
>
> Directly tuning the full model with first-order methods, however, is computationally expensive. To address this, we considered using a zeroth-order (ZO) method, which allows tuning without direct gradient calculations. But ZO tuning alone typically relies on hard prompts, which can be suboptimal for fine-tuning. For single-task scenarios, tuning the PEFT module alongside ZO tuning of the full model helps overcome these limitations, enhancing performance compared to using either ZO or PEFT alone. Our experimental results (Table 1 and Table 2) confirm this hypothesis.
>
> ## 3- Computational Cost of First-Order and Efficiency Comparison
>
> Please refer to the global responses on memory profiling and efficiency analysis, as well as Figure 5 and Table 6 in the revised version of our paper. Figure 5 demonstrates that computing first-order (FO) gradients for PEFT parameters is not significantly demanding in terms of memory usage. In fact, in some cases, zero-order (ZO) methods require more memory than FO PEFT. Also, our method is competitive to MeZO in terms of memory requirements, while addressing its limitations and converging much faster. Our method outperform MeZO by a large gap in half the number of iterations.
>
> We calculate the gradient of PEFT parameters because using ZO for both the full model parameters and the PEFT parameters results in higher approximation errors. Our goal is to achieve optimal performance **within the memory budget**. Calculating the exact gradient of PEFT parameters is feasible since it is significantly less expensive than FO full fine-tuning. Therefore, we choose First-Order PEFT fine-tuning to enhance performance.
>
> Regardless, if the memory constraints are extreme, our bilevel framework can incorporate doing ZO finetuning for updating the PEFT parameters too.

---

> > ### Comment · Reviewer_mLL4 · 2024-11-27
> > **Thanks for the rebuttal.**
> >
> > I appreciate the authors' response. I have further questions as below:
> >
> > 1. I still think introducing FO to ZO algorithm just in order to solve the hard prompts is neither very intuitive nor practical. First of all, the hard prompt issue itself is not so problematic, as only one set of hard prompt can be used for the entire dataset. Second, as a loss, one has to sacrafice the memory consumption to integrate FO, as indicated by the global response of the authors, that the memory consumption of the proposed algorithm = max(ZO, FO).
> >
> > 2. In the global response on the memory profiling, why in some scenarios, FO consumes even less memory than ZO? I guess even if this is correct, this would require some additional explanations.
> >
> > At the same time, I understand the authors' response but given the systematic design and the algorithm proposed in this work, I would keep my original score.

---

> > > ### Author Response · Authors · 2024-12-03
> > > **Final Response**
> > >
> > > Once again, we appreciate the reviewer’s feedback and have addressed all concerns raised. We have clarified that **the bilevel structure is responsible for mitigating the hard prompt sensitivity issue not FO**. Additionally, the flexibility of our bilevel framework, which allows the choice between ZO and FO depending on memory constraints, is a key strength.
> > > The differences in memory consumption between FO and ZO are clarified, with our method **consistently outperforming both while using as much memory as them**. We respectfully ask the reviewer to reconsider their score, as these points have been thoroughly addressed and clarified.

---

> ### Author Response · Authors · 2024-11-25
> **A gentle reminder**
>
> We greatly appreciate the time and effort you put into providing constructive feedback. Following your suggestions, we have carefully revised the manuscript to address your concerns. The updated version incorporates all recommended changes and improvements.
> We kindly request that you review the revised submission and our rebuttal and consider providing additional feedback. We would be especially grateful if you might consider adjusting your score in light of the improvements made.
> Thank you once again for your valuable input and for contributing to the quality of our work.

---

> ### Author Response · Authors · 2024-11-28
>
> We appreciate the reviewer's continued feedback. We address your concerns as follows:
>
> ### **Regarding the memory consumption of FO vs. ZO**
>
> The reason FO sometimes uses less memory than ZO is related to the nature of PEFT (Parameter-Efficient Fine-Tuning). In PEFT, the base model remains frozen, so we do not need to store the gradients of the base model during FO computations. On the other hand, ZO, while avoiding backpropagation, still requires storing the gradient estimates for the base model, leading to comparable or even higher memory consumption in certain scenarios. Importantly, the higher memory in certain scenarios depends on the number of tokens in a task sample on average. For instance, in datasets with a large number of tokens per task (e.g., MultiRC, used for profiling MEZO in their paper), MEZO performs better in terms of memory. However, this is not always the case, as the memory consumption is task-dependent.
>
>
> We would also like to address the reviewer’s concern, ‘Even if this is correct.’ Our claims are fully supported by evidence, and the code for MEZO and ZO-BENCH is publicly available. To further clarify, the memory profiling results can be independently verified, and we encourage you to explore the experiments to confirm these findings.
>
>
> ### **On the memory consumption being a merit**
>
> The observation that the memory consumption of our proposed algorithm is equivalent to the maximum of ZO and FO is, in fact, a **strength** of our method rather than a drawback. Our systematic design effectively leverages this characteristic to improve performance, enhancing either FO or ZO without exceeding their individual memory consumption limits. The algorithm’s performance is clearly very effective in largely improving the current methods for **both** single task and multi-task learning.
>
> ### **Hard prompt issue in ZO is problematic**
>
> Contrary to the reviewer’s comment that "the hard prompt issue itself is not so problematic," we would like to remind the reviewer that previous work has demonstrated that improper hard prompts can significantly impact performance of ZO.  This is why we proposed tuning the prompts instead of relying on hard prompts. Below is a detailed comparison between results with and without prompts:
>
> First, similar to Table 5 of the MeZO paper, we experimented with different choices of hard prompts. Table 1 below shows the MeZO results for tuning OPT 1.3b on SST-2 and COPA.
>
> ### Tab. 1: MeZO's Prompt Sensitivity
>
> | MeZO Experiment | With Prompt (\%) | Without Prompt (\%) |Difference|
> |-----------------|-----------------|----------------------|----------------------|
> | SST-2           | 89.6            | 51.9                 | -38.6|
> | COPA            | 70.0            | 54.8                 | -15.2 |
>
> We compare this with our method on OPT1.3b. Tab. 2 below summarizes our results:
>
> ### Tab. 2: Bilevel-ZOFO's Prompt Sensitivity
> | Bilevel-ZOFO Experiment | With Prompt (\%) | Without Prompt (\%) |Difference|
> |------------|-----------------|---------------------|---------------------|
> | SST-2      | 93.3            | 92.9                | -0.4 |
> | Copa       | 76.66           | 73.6                | -3.06|
>
> We can see that our method effectively mitigates the sensitivity of MeZO to hard prompts. The difference between the results with and without a simple hard prompt in our experiment is much smaller than MeZO's.
>
> We must clarify that **FO is not intended to mitigate the hard prompt issue; the bilevel structure is.** The bilevel framework we propose can flexibly use either ZO or FO for PEFT, depending on memory restrictions. This flexibility is one of the key merits of our contribution. As we explained previously, we choose to use FO because it provides better accuracy within the memory budget.
>
> **To improve clarity, we will add the above explanations to the main text. Based on these points, we respectfully encourage you to consider this contribution and reconsider your score. If you have any additional questions or require further clarification, we are happy to provide further explanations.**

---

### Official Review · Reviewer_DSjq · 2024-11-03

**Soundness:** 3
**Presentation:** 2
**Contribution:** 2
**Rating:** 6
**Confidence:** 4

**Summary:**

This paper realised two issues in current LLM tuning research. One is that parameter-efficient fine-tuning (PEFT) cannot sufficiently compete with full-model tuning, because only part of model parameters are tuned, which limits the model capacity. The second is that zeroth-order (ZO) optimization can tune full model parameters while relying on fixed non-optimal prompts. The authors propose to complement ZO with PEFT to mitigate the sensitivity to hard prompts through bi-level optimization. During formulation, they first transform the objective into single-level optimization problem and then use zeroth-order information to approximate gradient.

**Strengths:**

1. This paper presents a bi-level optimization method that is more suitable for tuning full pre-trained large language models, compared with parameter efficient tuning and zeroth-order tuning.
2. The proposed method can be extended to a lightweight meta-training process for multi-task learning.
3. The final results outperform both FO.and recent work MeZO.

**Weaknesses:**

1. I understand the motivation of complementing ZO with PEFT, but bi-level is known for its high computation cost, even after the zeroth-order approximation. In this sense, how is the tuning efficiency of the proposed method?
2.On line 201, you split data into two parts. Can the two-level optimization share the same tuning dataset?
3. From Eq.3 to Eq. 7, this process is following existing works. Any new technical contributions here? And have you demonstrated this transformation is better than others?
4. Efficiency comparision is lacked in experiments.
5. There are some improved LLM tuning works over MeZO. If this is not necessary to compare or even mention them, please provide the reasons.
6. Why not consider tuning soft prompts in this paper? As zeroth-order techniques are also explicitly used to tune prompts. [1]

[1] Black-Box Tuning for Language-Model-as-a-Service. ICML 2022.

**Questions:**

Please refer to Weaknesses.

The theoretical analysis presents the convergence of bi-level optimisation. It might be interesting for some readers but I doubt its significance in LLM tuning. For example, can you explain how will the convergence theory instruct the tuning process in real applications?

It would be better to draw a pipeline to show the optimization procedure, which can be left to appendix if not sufficient space.

---

> ### Author Response · Authors · 2024-11-21
> **Response to Reviewer DSjq**
>
> We thank the reviewer for thier valuable comments and feedback. We address their concerns below:
>
> ## 1- Efficiency of the proposed bilevel method
> We now compare the memory requirement and the wall-clock-time efficiency of our method to the baseline. (see the global response and  Table 6 and Figure 5 now added to the appendix of the revised version of the paper), our method is as computationally efficient as the baselines.
>
> We also mentioned in the method section (lines 216-221) that traditional bilevel optimization methods tend to be computationally intensive, primarily due to the need for multiple Hessian-vector products. However, recent advances have introduced a new approach to solving bilevel problems that requires only first-order information, significantly reducing computational costs. Our method aligns with this newer approach and we further reduce the computational cost using the zeroth order approximation of the base model. This allows us to achieve more efficient computation.
>
> ## 2- Can the two-level optimizations share the same tuning dataset?
>
> In this work, we consider two settings: single-task fine-tuning and multitask learning. For single-task fine-tuning, PEFT models (such as prompts) can be treated as hyperparameters for the full model. Typically, we use a separate dataset for hyperparameter tuning to avoid overfitting the data used in the upper level.
>
> In multitask learning, especially in few-shot learning, the lower level focuses on training task-specific parameters, while the upper level trains a meta-model that generalizes well across all tasks. After meta-training, the PEFT model can be further trained on a few new data points when applied to an unseen task. Our bilevel training process is designed to simulate this application process, which is why the datasets used in the lower and upper levels are distinct.
>
>
> ## 3- Any new technical contributions From Eq. 3 to Eq. 7?
>
> The technical challenge in this work lies in incorporating zeroth-order gradient estimation into a penalty-based bilevel optimization framework. Most existing studies on bilevel optimization focus on providing theoretical convergence guarantees for first-order methods. Given the nested structure of the upper and lower-level problems, it is not immediately clear whether a zeroth-order approach would achieve both theoretical and empirical convergence.
>
> In this work, we demonstrate that our proposed method not only performs effectively but also has theoretical convergence guarantees. For LLM applications, the contribution of this work is a more effective framework to advancing zeroth-order fine-tuning and enhancing multitask learning.
>
> ## 4- Lack of efficiency comparisons
>
> Please refer to the global responses for the efficiency comparison. We have now added memory profiling and wall-clock-time analysis.
>
> ## 5- Comparisons with improvements over MeZO
> We thank the reviewer for pointing this out. We are aware of other methods to improve MeZO, as noted in the related work section (please let us know if we missed any). Since our proposed structure is **orthogonal** the choice of the zeroth-order tuning methods, our method can integrate other zeroth-order variations in the upper-level steps. However, due to time and resource limitations—and, more importantly, to maintain focus on the bilevel structure and its unique multitask learning capabilities—we used the classic MeZO as a baseline to demonstrate the effectiveness of our approach.
>
>
> ## 6- Why not consider tuning soft prompts in this paper?
> Thank you for your question. In our lower level, one of our PEFT models involves tuning the soft prompt. Unlike [1], we use a first-order method for tuning prompts. We didn't use a zeroth-order approach because soft prompts have significantly fewer trainable parameters than the full pre-trained model, making it computationally inexpensive to calculate their exact gradient as opposed to tuning the whole model. This allows for more accurate estimation compared to zeroth-order methods.
>
> The setting in [1] differs from ours. [1] addresses a black-box scenario where access to the full model parameters is restricted, whereas we consider a white-box scenario where we have full access to the model parameters, enabling zeroth order fine-tuning within pre-trained LLMs. We have now included [1] in the second paragraph on page 1 when discussing soft prompt tuning.
>
> ## 7- Question about the theoretical analysis
> Thank you for your comment. We included the theoretical convergence analysis because bilevel optimization is challenging to train due to its nested structure between the upper and lower level problems. Given this complexity, it is not immediately obvious that the new method incorporated with zeroth order approximation will work effectively in practice. Our theoretical analysis ensures that the proposed method will not diverge when applied in experiments, providing a foundation of reliability for real applications.

---

> > ### Author Response · Authors · 2024-11-21
> >
> > ## 8- Optimization Pipeline Figure
> > Thank you for the suggestion. We have now added a pipeline illustration in the revised version. Please refer to Figure 3 in the appendix (page 20).

---

> > > ### Comment · Reviewer_DSjq · 2024-11-21
> > >
> > > Thanks for the response. Although some points (e.g., 1 and 3) are not very convincing to me from the view of optimization, most of my concerns have been addressed. I think the revised version should be good for LLMs tuning tasks. Thus, I would like to raise my score.

---

> > > > ### Author Response · Authors · 2024-11-21
> > > >
> > > > We sincerely appreciate your review and the increase in your score. Your feedback has been very helpful in improving our work. Thank you for your time and careful consideration of our work.

---

### Official Review · Reviewer_21GX · 2024-11-04

**Soundness:** 2
**Presentation:** 2
**Contribution:** 2
**Rating:** 5
**Confidence:** 3

**Summary:**

This paper proposes a bilevel optimization framework and combines zero-order finetuning with first-order PEFT to achieve higher performance on the downstream task. Experiments show that their proposed method outperformance both zero-order finetuning and PEFT only.

**Strengths:**

1. The proposed method is novel and interesting.
2. The experiment results look pretty strong compared to vanilla finetuning.

**Weaknesses:**

1. The motivation of this paper is unclear. The authors mention that with the bilevel optimization, the sensitivity of ZO to hard prompts can be mitigated. Are there any evidence for this claim? Or is the performance improvement simply comes from more tunable parameters?
2. Similar to point 1, have the authors done experiments with PEFT followed by a ZO finetuning?
3. The experiment section is a bit unclear, the MeZO baseline is not explained clearly.

**Questions:**

Please see the weaknesses above.

---

> ### Author Response · Authors · 2024-11-21
> **Response to Reviewer 21GX**
>
> We sincerely thank the reviewer for their feedback. Below we address the concerns raised:
>
> ## 1- Mitigating the Sensitivity of ZO
>
> We thank the reviewer for mentioning this. We now have added more experiments to show that our method mitigates the sensitivity of MeZO to hard prompts.
>
> ### 1.1- Sensitivity to different hard prompts
> First, similar to Table 5 of the MeZO paper, we experiment with different choices of hard prompts. Tab. 1 below shows the MeZO results for tuning Opt 1.3b on SST2 and COPA.
>
> ### Tab. 1: MeZO's Prompt Sensitivity
>
> | MeZO Experiment | With Prompt (\%) | Without Prompt (\%) |Difference|
> |-----------------|-----------------|----------------------|----------------------|
> | SST-2           | 89.6            | 51.9                 | -38.6|
> | COPA            | 70.0            | 54.8                 | -15.2 |
>
>
>
> We compare this with our method on OPT1.3b. Tab. 2 below summarizes our results:
>
> ### Tab. 2: Bilevel-ZOFO's Prompt Sensitivity
> | Bilevel-ZOFO Experiment | With Prompt (\%) | Without Prompt (\%) |Difference|
> |------------|-----------------|---------------------|---------------------|
> | SST-2      | 93.3            | 92.9                | -0.4 |
> | Copa       | 76.66           | 73.6                | -3.06|
>
> We can see that our method effectively mitigates the sensitivity of MeZO to hard prompts. The difference between the results with and without a simple hard prompt in our experiment is much less than MeZO's
>
> ### 1.2- The effectiveness is not the result of more parameters (Experiments of PEFT followed by ZO finetuning)
>
> To also validate that the improved results are not because of tuning more parameters, we conducted an experiment on COPA using OPT1.3B.
>
> **Two-stage tuning**: First, we performed first-order prompt tuning for a fixed number of steps (same as the number of lower-level updates in bilevel-ZOFO), followed by additional tuning using ZO for the same number of iterations as the upper level updates in bilevel-ZOFO:
>
> As shown in our experiments (see Tab. 3 below), the two-stage method does not does not work at all because the accuracy after a first stage is 74.33, while at the end of the two-stage setting the accuracy becomes 51.66. Our bilevel structure makes the trained prompts dynamically optimal for the full ZO fine-tuning.
>
> ### Tab. 3:
> |     Method          | COPA Stage 1 (Prompt Tuning with FO) | COPA Stage 2 (Fine Tuning with ZO) | Ours|
> |---------------|--------------------------------------|------------------------------------|-----|
> | Performance          | 74.33                                | 51.66                              |76.66|
>
>
>
> ## 3- Unclear MeZO baseline
>
> We sincerely thank the reviewer for this valuable feedback and apologize for the confusion.  MeZO, introduced by Malladi et al., is the first method to apply ZO fine-tuning to LLMs. More specifically, MeZO is replacing the gradient in the model with the approximation (Equation 7 in our paper) and then doing SGD or Adam. We have revised the paper to clearly define and explain what MeZO refers to, please see the first paragraph in Section 2.1 as well as the second paraphraph in Section 4 in the revised version. Moreover, we have now included all details of the experimental setting in Appendix B.
>
> ---
>
> Once again, we appreciate your constructive and thorough comments. We hope our explanations have addressed your concerns.

---

> ### Author Response · Authors · 2024-11-25
> **A gentle reminder**
>
> We greatly appreciate the time and effort you put into providing constructive feedback. Following your suggestions, we have carefully revised the manuscript to address your concerns. The updated version incorporates all recommended changes and improvements.
> We kindly request that you review the revised submission and our rebuttal and consider providing additional feedback. We would be especially grateful if you might consider adjusting your score in light of the improvements made.
> Thank you once again for your valuable input and for contributing to the quality of our work.

---

> ### Comment · Reviewer_21GX · 2024-11-30
>
> Thanks the authors for the additional experiments. However I still have some questions on the additional experiment results.
> For Tab. 3, it's not reasonable that the performance of stage 2 is much lower than the result from stage 1. The author should conduct basic hyperparameter tuning to make sure the result is reasonable or explain the counter-intuitive result.
>
> Now I understand the MeZO baseline, but it seems the existence of MeZO weakens the novelty of this work. (This should be a minor point though).
>
> Therefore, I'll keep my rating.

---

> ### Author Response · Authors · 2024-12-01
>
> Thank you for your continued feedback and for highlighting the importance of including additional experiments to further support our claims.
> ## 1. Regarding Tab. 3 and the Counter-Intuitive Result in Stage 2
>
> To address your concern, we conducted additional hyperparameter tuning to ensure our reported results accurately reflect that two-stage training is not effective. The updated results are derived from the same range of hyperparameters used in Bilevel-ZOFO as reported in Section B.1.3 of the appendix. We initially tested using only the best hyperparameters identified from the results in Table 1 of the paper. However, following the reviewer's suggestion, we re-ran the experiments with all hyperparameters. These results represent the mean of three seeds:
>
> | **Accuracy after Stage I** | 74.33333 |
> |----------------------------|----------|
> | **Accuracy after Stage II**|          |
> | lr 0.001                   | 51.66667 |
> | lr 0.0001                  | 70.33333 |
> | lr 0.00001                 | 72.66667 |
> | lr 0.000001                | 74.33333 |
> |----------------------------|----------|
> | **Ours**                   | 76.6666  |
>
> The results indicate the following:
>
> 1. Even with extensive hyperparameter tuning, the second stage does not improve the results achieved after the first stage and is highly likely to **decrease** performance.
> 2. Our method, however, **improves performance** when using **the same number of steps** in the upper and lower levels, respectively.
>
> The observed performance drop after the second stage is indeed counter-intuitive at first glance. However, it is a limitation of MeZO as it approximates gradients. While further fine-tuning intuitively should improve performance, the inherent noise in gradient approximation can lead to suboptimal updates. This observation is consistent with the fact that MeZO typically requires a significant number of iterations to converge. This is a key contribution of our work: **Our approach addresses MeZO's challenges**, such as sensitivity to hard prompts and long convergence times, while outperforming both MeZO and PEFT and maintaining similar memory efficiency.
>
> The intuition behind why our method is effective in enhancing both MeZO's full-model tuning and PEFT is in the nested bilevel structure. This structure encodes more information (as reflected in the training method) from the prompt tuning stage than only treating it as a first stage, thereby providing better guidance for MeZO.
>
> We will also include an expanded explanation and additional experimental results in the manuscript to clarify the difference between our method and the two-stage strategy.
>
>
>
>
> ## 2. Regarding the Novelty in the Presence of MeZO:
>
> We would like to clarify that the novelty and contributions of our work build upon and extend MeZO.
>
> - **More Robust:** We address MeZO’s limitations by mitigating its sensitivity to hard prompts, as demonstrated in Tables 1 and 2 in the previous rebuttal.
>
> - **More Efficient and Effective Multitask Training Method:** Compared to previous methods for multitask training, our approach is lightweight while achieving better performance. These improvements highlight our method's unique contribution to enhancing efficiency and scalability.
>
> - **Faster Convergence:** Our approach significantly reduces the number of iterations required by MeZO, enabling faster and more reliable convergence.
>
> - **Optimizing the Merits of MeZO and PEFT:** The proposed nested bilevel structure enhances the performance of both MeZO’s full-model tuning and PEFT. Our method outperforms MeZO and PEFT while **maintaining similar memory efficiency.**
>
> We hope these additional experiments and explanations regarding our contributions address your concerns. If they resolve your concerns, we kindly ask for a reconsideration of the rating. If any additional issues remain, we would be happy to provide further clarifications.

---

> > ### Comment · Reviewer_21GX · 2024-12-01
> >
> > Thanks to the authors for additional results. However the updated Tab.3 is still not convincing to me. I understand that MeZO can have suboptimal gradient estimation. But it's still not reasonable that MeZO after PEFT can only decrease the performance. Does it mean that the training loss never decreases when running MeZO at the second stage? Or is the worse performance from overfitting to the training set? Can the authors show the training curve of second-stage MeZO?

---

> ### Author Response · Authors · 2024-12-02
>
> Thank you for your continued feedback. We appreciate the opportunity to provide further clarification.
>
> The training loss curves for both stages of a two-stage approach and our bilevel framework are provided at the following links for reference:
> - First-stage PEFT training loss: [link](https://postimg.cc/G97Tf2Nq)
> - Second-stage MeZO training loss: [link](https://postimg.cc/Y4zhzqYN)
> - Our bilevel method training loss: [link](https://postimg.cc/WqhzVGKS)
>
> When running MeZO in the second stage, the training loss exhibits oscillations and does not show improvement within 500–1000 iterations. This behavior is consistent with findings in the original MeZO paper, which notes that MeZO typically requires much longer to converge—on the order of 100k iterations. The oscillatory behavior observed within the shorter training duration is not surprising due to gradient approximation errors.
>
> In contrast, our bilevel method effectively addresses the issues of MeZO and demonstrates improved performance over both MeZO and the PEFT baseline, even **with the same number of ZO iterations**.
>
> We hope this additional clarification and the provided evidence address your concerns.

---

> ### Comment · Reviewer_21GX · 2024-12-02
>
> Thanks the authors for additional experiments. I'm more convinced about the reported results. I suggest the authors to include more comprehensive experiments in the revision. I've also lower my confidence as I'm not familiar with the MeZO baseline.
>
> And another question about the experiments. Why do the authors choose to evaluate under a low-resource setting where the training set only contains 1000 examples? I understand this is an additional question not included in the initial review, but I feel this is important for evaluating this work.

---

> ### Author Response · Authors · 2024-12-02
>
> Thank you for your thoughtful feedback and for taking the time to share your insights. We're glad our explanation addressed your concerns and appreciate your consideration of our additional experiments.
>
> Regarding your question about the choice of evaluating under a low-resource setting with only 1000 training examples: This work builds upon MeZO, which served as the foundation and baseline for our work. For single-task experiments, we aligned the number of data points with MeZO (See Appendix E of their paper) to ensure a direct and fair comparison, facilitating straightforward interpretation of improvements for readers.  Similarly, in the multi-task setting, we based our sample distribution on MetaICL, which serves as another critical baseline for our work.
>
> By adhering to these established benchmarks, we aim to provide a clear and consistent basis for comparison between our work and the respective baselines.
>
> We will also take your suggestion of including more comprehensive experiments into account for our revision to further strengthen the work.
>
> Thank you again for your valuable comments and suggestions. We hope you would consider increasing your score, as it will greatly support the impact of this work. If you have further concerns, we will be more than happy to address them.

---

> > ### Comment · Reviewer_21GX · 2024-12-02
> >
> > Thanks for the follow-up responses.
> > I'm hesitating with increasing the score as
> > 1. I'm not very familiar with related works. I've lowered my confidence to reflect this.
> > 2. Although the performance is strong compared to vanilla MeZO, performance improvement compared to FO PEFT is not that significant. The results shown in Table 3 about Multi-task Meta learning are also a bit mixed, especially for the row of Ours (Zero-shot). The authors should also explain the difference between Ours(Zero-shot) and Ours(tuned) more clearly.
> >
> > Therefore, I'm still not confident enough to support acceptance of this paper.

---

> > > ### Author Response · Authors · 2024-12-02
> > >
> > > Thank you for your continued feedback. While we appreciate the thoroughness of the review process, we are disappointed by the conclusion that the results are not significant or that unfamiliarity with MeZO could justify hesitation in supporting this work.
> > >
> > > Throughout the discussion period, we have diligently addressed all concerns raised, providing detailed responses, new experiments, and evidence to substantiate our claims. We have outlined MeZO and its limitations, explained the motivation behind our work to address these issues, and highlighted the improvements our method brings. We have demonstrated our method **effectively addresses MeZO's sensitivity to the choice of hard prompts**, reducing the accuracy drop from -38.6 and -15.2 to -0.4 and -3.1. Furthermore, this work **achieves faster convergence**, and **delivers superior performance** compared to MeZO. Importantly, these advancements are not the result of tuning additional parameters, as we explicitly showed that our bilevel optimization is necessary for improving beyond standard PEFT methods, which a simple two-stage PEFT-MeZO approach fails to achieve.
> > >
> > > Moreover, the claim that performance improvement over FO PEFT is "not significant" does not align with the evidence provided. As detailed in Tables 1 and 2, the reported results are **averages across multiple random seeds and extensive hyperparameter searches, not isolated cases or cherry-picked results**. Our method consistently outperforms PEFT methods across diverse tasks, model sizes, and experimental conditions. Given the established status of PEFT methods in both research and industry, outperforming them across the board is indeed a significant result.
> > >
> > > ## Regarding Table 3 and the distinction between Ours (Zero-shot) and Ours (Tuned):
> > > Thank you for pointing this out. We have addressed this in the paper (lines 419–424). To ensure clarity, we will revise the discussion of the results in the paper to more explicitly explain the distinctions between Ours (Zero-shot) and Ours (Tuned).
> > >
> > > To clarify: The difference lies in how we evaluate the obtained model. Specifically, *Ours (Zero-shot)* refers to evaluating our method without any demonstration examples or any additional fine-tuning. Unlike ICL methods, which append demonstration examples to every test input—an approach that can be computationally intensive during inference—we do not include demonstration examples during evaluation (Hence the term zero-shot). In contrast, *Ours (Tuned)* involves additional LoRA fine-tuning our method on these demonstration examples (using the same number of demonstration examples required by ICL). In summary:
> > >
> > > - Zero-Shot: no train ---> evaluate with no fine-tuning and no demonstration samples.
> > > - Few Shot (ICL): no train ---> evaluate with appending demonstration samples to every test input.
> > > - MetaICL: meta train on training data + appending additional demonstration examples to every training sample (more training time) ---> evaluate with appending demonstration samples to every test input.
> > > - Ours(zero-shot): meta train on training data ---> evaluate with no fine-tuning and no demonstration samples.
> > > - Ours(tuned): meta train on training data ---> LoRA fine tune on demonstration samples---> evaluate with no demonstration samples (inference time efficient compared to ICL and MetaICL).
> > >
> > > We will add this pipeline to the revised version of the paper.
> > >
> > > Our approach highlights the flexibility and efficiency of our method, as it can adapt to unseen tasks with minimal examples, a brief fine-tuning phase, and ultimately requires significantly fewer tokens per test sample in inference.  This balance between computational efficiency and performance is a key spirit of our work.
> > >
> > > We believe this work **makes meaningful advancements** by **addressing the limitations of MeZO**, **outperforming standard PEFT methods**, and **introducing novel capabilities in the multi-task and few-shot settings**. The unfamiliarity with MeZO should not detract from the merit of this work, especially given the detailed explanations and evidence we have provided. **We respectfully request that you reconsider your evaluation and scoring, as the concerns raised have been thoroughly addressed, and this work makes notable contributions to the field.**

---

> > > ### Author Response · Authors · 2024-12-03
> > >
> > > We appreciate the reviewer’s continued feedback throughout this period . We have carefully addressed all concerns, providing new experiments and evidence that demonstrate our method **effectively mitigates MeZO’s sensitivity to hard prompts and outperforms it with faster convergence and superior performance**. Our results are **significant and are based on extensive hyper-parameter searches and multiple random seeds**, consistently outperforming PEFT methods. Additionally, we clarified the distinction between "Zero-shot" and "Tuned" versions of our approach. We respectfully ask the reviewer to reconsider their evaluation, as we have addressed all concerns.

---

### Official Review · Reviewer_tCaA · 2024-11-06

**Soundness:** 2
**Presentation:** 2
**Contribution:** 2
**Rating:** 6
**Confidence:** 3

**Summary:**

The paper propose  Bilevel ZOFO, a framework combining PEFT and Zeroth-Order (ZO) optimization to improve fine-tuning and meta-training efficiency for large language models. Using a bilevel approach, it achieves better performance in single- and multi-task settings.
It provides convergence guarantees for the optimization process.

**Strengths:**

Bilevel ZOFO effectively combines PEFT and ZO optimization to reduce computational and memory costs, making large model fine-tuning more efficient.
Its bilevel structure allows for competitive performance in both single- and multi-task settings with theoretical convergence guarantees.

**Weaknesses:**

The experiments are limited to small- to medium-sized models in its experiments, raising concerns about scalability to larger models commonly used in practice and fair comparison.

**Questions:**

Q1 Could you provide quantitative evaluations for convergence, wall-clock time per step, memory profiling, and memory consumption, as seen in the MeZO paper?

Unlike MeZO that only require forward passes,  Bilevel ZOFO relies on First-Order gradient calculations, which can significantly increase computational and memory demands.
Minimax optimization often faces convergence problems due to saddle points, which can cause the training to stall and slow down overall progress.


Q2 Could you provide the experimental results for larger models such as OPT13B.

ZO methods are particularly beneficial for reducing computational resources in large-scale model training. Thus, existing methods have generally evaluated models of at least moderate scale, such as models larger than OPT13B. The experiments in this paper, however, are limited to small- to medium-sized  models, up to 7B parameters.
To ensure a fair comparison with existing methods, it is essential to test on models of at least comparable size, such as OPT13B.
Effective hyperparameters may vary significantly based on model size.
However, MeZO tested in this paper may not necessarily be optimal for small-scale settings.
as existing methods are not designed with small-scale models.


Q3 Could you provide detailed experimental settings?
The MeZO paper provides detailed information on hyperparameters and computational environment, which is necessary for reproducibility and accurate comparison.

---

> ### Author Response · Authors · 2024-11-21
> **Response to Reviewer tCaA**
>
> We thank Reviewer tCaA for their valuable feedback. Below we address the concerns raised by the reviewer.
>
> ## 1- Larger Scale Models
> We thank the reviewer for highlighting this concern.
>
> While we recognize that scaling to much larger models for single-task experiments, such as OPT13B, would further strengthen our work, our testing was limited by resource constraints.  As with most academic labs, we can afford and focus on researching 1B-7B parameter models, which should be sufficient for testing and validating research ideas.
>
> Our experiments have already demonstrated the effectiveness of our methods on models ranging from 1B to 7B parameters. Similarly, MeZO has shown strong performance on 1B-7B models, as reported in the original paper. Our findings are consistent with MeZO, and Bilevel-ZOFO consistently outperforms other baseline methods across this model range. Based on these results, we believe our method would yield similar improvements on even larger models.
>
> Nonetheless, we are actively working to extend our method to larger-scale models and will share those results as soon as they are available.
>
>
> ## 2- Memory Profiling and Wall Clock Analysis
>
> Please see the global comment. We have now included the memory profiling and wall-clock-time analysis of our method compared to the baselines.
>
> ## 3- Quantitative Evaluations for Convergence
>  We now added Figure 4 in the revised version to show that Bilevel-ZOFO converges smoothly. Figure 4 presents the training loss for the lower-level objective of the bilevel framework with Lora as the PEFT model. As shown, consistent with the guarantees provided by our theoretical analysis, Bilevel-ZOFO converges.
>
> ## 4-Experimental Setting and Hyperparameters
>
> We now have provided all details on hyperparameters and our computational environment in **Appendix B** of the revised paper and include them here for your reference:
>
> **Training Data:**
> For each task, we use 1,000 training examples, 500 validation samples, and 1,000 test samples. In Bilevel-ZOFO experiments, the training data is split into upper-level and lower-level subsets using a 1:2 ratio.
>
> **Hyperparameter Search:**
> * For FO PEFT and MeZO, we explored learning rates from ${1e-2, 1e-3, 1e-4, 1e-5, 1e-6}$ across all variants. The baseline hyperparameters are consistent with those used in the MeZO experiments. We also set $\epsilon=0.001$, following MeZO.
>
> * For Bilevel-ZOFO, we tuned both upper- and lower-level learning rates with {1e-4, 1e-5, 1e-6, 1e-7} and {1e-2, 1e-3, 1e-4, 1e-5}, respectively.
>
> **Training Details:**
>
> We use Adam optimizer with betas=(0.9, 0.999) for the baselines and for both upper and lower updates of our method.
>
> All experiments used a batch size of 8 and were conducted in bfloat16 precision on a single A6000 Ada 48GB GPU. MeZO was run for 10,000 steps, while FO and Bilevel-ZOFO methods were run for 5,000 steps. Our implementation builds upon MeZO’s codebase, and memory profiling as well as latency calculations are based on their framework.
>
> If there is any details missing please point them out, and we would be happy to provide additional information. We thank you again for your feedback.

---

> ### Author Response · Authors · 2024-11-25
> **A gentle Reminder**
>
> We greatly appreciate the time and effort you put into providing constructive feedback. Following your suggestions, we have carefully revised the manuscript to address your concerns. The updated version incorporates all recommended changes and improvements.
> We kindly request that you review the revised submission and our rebuttal and consider providing additional feedback. We would be especially grateful if you might consider adjusting your score in light of the improvements made.
> Thank you once again for your valuable input and for contributing to the quality of our work.

---

> > ### Comment · Reviewer_tCaA · 2024-11-30
> >
> > Thank you for your response. My concerns have been resolved, so I have raised the score.

---

> > > ### Author Response · Authors · 2024-12-01
> > >
> > > Thank you for taking the time to review our responses and update your score. We truly appreciate your thoughtful consideration. If there are any additional concerns or questions, we would be happy to provide further clarification.

---

### Author Response · Authors · 2024-11-21
**Global Comment Addressing Shared Concern**

We thank the reviewers for their efforts and valuable feedback.  As there are is common concern among the reviewers about lack of memory profiling and wall-clock-time analysis, we address it here.

In our experimental settings, the memory usage of our method matches that of the most memory-intensive baselines. We have now included **Figure 5** and **Table 6** in the revised version (pages 23 and 24 in the appendix) to address these concerns more comprehensively. Our implementation builds upon MeZO’s codebase, and memory profiling as well as latency calculations are based on their code. We summarize the results below.

## **Memory Profiling:**
Figure 5 (now included in the paper) illustrates memory consumption across tasks for Bilevel-ZOFO, MeZO, and FO Prefix-Tuning. Memory usage varies depending on the task, with MeZO and first-order PEFT methods occasionally surpassing each other. The GPU memory requirement of our method is comparable to the memory usage of FO PEFT and MeZO experiments. Despite this, as demonstrated in Table 1 of the paper, our method consistently outperforms both PEFT and MeZO on average. We include the values of Figure 5 in Tab. 1 below for ease of reference. Specifically, the values correspond to maximum GPU memory allocated in terms for GB for  fine-tuning OPT1.3b on each task using a batch size of 8 and on a single A6000ada 48GB GPU.

### Tab. 1: Memory Consumption Comparison (Values from Figure 5 of the paper)

| Datasets | MeZO  | FO (Prefix) | Bilevel-ZOFO (Prefix)|
|----------|-----------|------------------|----------------------------|
| Copa     | 8.12      | 2.74             | 8.12                       |
| MultiRC  | 10.12     | 22.72            | 22.72                      |
| ReCoRD   | 8.99      | 11.22            | 11.22                      |
| WSC      | 8.13      | 4.24             | 8.13                       |

## **Wall-clock Time Analysis**

We also present a wall-clock time analysis of bilevel-ZOFO compared to the baseline. Tab. 2 below shows the results (See Table 6 in the appendix for a complete version). Similar to MeZO, we observe that each zeroth-order tuning steps takes longer compared to first-order steps. The results indicate that our bilevel-ZOFO achieves comparable step duration to the FO-PEFT method while significantly requiring fewer step duration compared to MeZO. Moreover, as highlighted in Table 1 of our paper, bilevel-ZOFO outperforms both methods on average.

The values in Tab.2 correspond to the duration of each step of the methods in seconds. The values are measured based on tuning OPT1.3b on a single A6000ada 48GB GPU. The wallclock time is averaged over 3 different runs that produced the values of Table 1 of our paper. We use a batch size of 8 for all experiments.
|

### Tab. 2: Wall clock time duration (in seconds) of each step of our method and the baselines. (Values from Table 6 of the paper)
Task     | MeZO | FO Prefix-Tuning  | Bilevel-ZOFO (Prefix) |
|----------|----------|----------------------|---------------------------|
| Copa     | 0.299    | 0.127                | 0.135                     |
| MultiRC  | 0.622    | 0.474                | 0.502                     |
| WSC      | 0.278    | 0.120                | 0.164                     |

---

### Author Response · Authors · 2024-12-04
**Summary of the discussion period**

We sincerely thank the Area Chair and the reviewers for their efforts. The discussion improved the clarity and overall presentation of our work. During this rebuttal process, the reviewers highlighted the following strengths in our work:

1. **Novelty:** This work introduces a novel bilevel optimization framework specifically designed for fine-tuning large pre-trained language models.

2. **Efficiency and Performance:** This work use bilevel structure that relates  PEFT to zeroth-order (ZO) optimization to significantly reduce computational and memory costs, achieving competitive results in both single-task and multi-task settings. It outperforms First-Order PEFT, and Zeroth-order approaches like MeZO.

3. **Multitask Learning:** This method supports lightweight meta-training processes, making it suitable for efficient multi-task learning in diverse applications.  The proposed method is particularly beneficial in resource-limited settings, such as medical domains with limited labeled data.

4. **Theoretical Soundness:** We provided theoretical guarantees for convergence, enhancing the framework's reliability.

They also raised some concerns which we summarize below and explain how they were addressed. There were minor concerns, such as correcting notation and including a pipeline figure that we fully addressed. However, we have omitted them here for the sake of brevity.

------------
## Questions, Concerns of Reviewer tCaA and How We Addressed Them:

1. **Scalability to models larger than 13B:** We demonstrated consistent performance across 1B–7B models and clarified that the method is expected to scale effectively to larger models, given the strong baseline performance in this range.


2. **Memory and Time Profiling:** We provided detailed analyses showing bilevel-ZOFO does not increase the memory usage of PEFT and MeZO, while outperforming both.


3. **Convergence guarantee beyond theory:** We added evaluations confirming smooth convergence aligned with theoretical guarantees.


4. **Experimental Details:** We expanded experimental settings, hyperparameters, and computational environment descriptions for clarity and comparability.

**Outcome:**
The reviewer acknowledged that all concerns were fully addressed and increased their score.

------------
## Questions, Concerns of Reviewer 21GX and How We Addressed Them:

1. **Motivation and Sensitivity of ZO to Hard Prompts:**


   - We added additional experiments to demonstrate that Bilevel-ZOFO effectively mitigates MeZO’s sensitivity to hard prompts, maintaining robust performance with minimal variance.


   - We validated that the improvements are not due to more tunable parameters through comparisons with a two-stage PEFT-MeZO approach, which showed degraded performance compared to Bilevel-ZOFO.


     - We conducted rigorous hyper-parameter tuning and presented training loss curves for two-stage PEFT-MeZO experiments as requested by the reviewer, confirming the limitations of a two-stage pipeline and justifying Bilevel-ZOFO’s effectiveness.

2. **Explanation of MeZO Baseline:**


   - We provided detailed explanations of MeZO’s methodology and limitations to enhance clarity in the manuscript and offer sufficient background for readers totally unfamiliar with the approach.

3. **Question about Low-resource setting Rationale**

   We addressed the reason for the low-resource experimental setting is to align with benchmarks from prior work (e.g., MeZO) to ensure fairness in comparison.

**Outcome:**

The reviewer acknowledged that their concerns were addressed but hesitated to increase their score. They reasoned that they are unfamiliar with related works and that our performance improvement over PEFT was not significant, despite having highlighted it as a strength in their review at the beginning.

We further emphasized the novelty and impact of the work, clarifying inconsistencies in the reviewer’s evaluation and providing additional evidence for the significance of the results. Despite we resolving all concerns, the reviewer did not further engage in discussion.

---

> ### Author Response · Authors · 2024-12-04
>
> ## Questions, Concerns of Reviewer DSjq and How We Addressed Them:
>
> 1. **Computational Cost Comparisons with Other Methods:** We included detailed efficiency comparisons, demonstrating competitive performance and reduced computational demands (Table 6, Figure 5). Additionally, we explained how the proposed bilevel method is not as expensive as classical bilevel methods.
>
> 2. **Using the Same Dataset for Both Levels of Optimization:** We explained that distinct datasets are used to avoid overfitting in single-task fine-tuning and to simulate few-shot multitask learning.
>
> 3. **Novelty in the Method and Contribution of Theoretical Analysis:** We elaborated on the theoretical and empirical difficulties of integrating ZO gradient estimation into bilevel optimization. We emphasized the importance of theoretically ensuring convergence and reliability in real-world applications.
>
> 4. **Why not Use Improved MeZO Variants:** We clarified that our method focuses on bilevel structures and multitask capabilities, making it orthogonal to the choice of zeroth-order tuning methods. Additionally, we highlighted that the proposed method is compatible with other ZO variants, facilitating future integrations.
>
> **Outcome:**
> The reviewer acknowledged the responses and the revisions, noting that their concerns were addressed and raising their score.
>
> -----
>
> ## Questions, Concerns of Reviewer mLL4 and How We Addressed Them:
>
> 1. **The need for simultaneous tuning full model and PEFT:** We explained that the bilevel structure with ZO for the full model and FO for PEFT enhances task-specific performance, supported by empirical results. We also showed that the memory usage of our approach does not exceed the memory usage of PEFT or MeZO.
>
> 2. **ZO motivation and memory profiling:** We added efficiency comparisons (Fig. 5, Table 6), showing our method balances memory and convergence faster than MeZO, with FO sometimes requiring less memory due to frozen base models.
>
> 3. **How this method mitigates Hard Prompt Sensitivity:** We provided experimental evidence showing significant sensitivity in MeZO versus significant robustness in our approach due to the bilevel structure.
>
> **Outcome:**
> We addressed all concerns with supportive evidence, including clarifications on notation, efficiency profiling, and detailed justifications for our method’s design choices. While the reviewer acknowledged our responses, **they believed that incorporating the FO method in our approach was intended to address the hard prompt issue**. However, we explained that this is incorrect --**the bilevel structure deals with  that aspect**. Instead, using FO in PEFT tuning in our approach ensures performance improvements within the memory budget. Despite us resolving all concerns, the reviewer did not further engage.

---

### Meta-Review · Area_Chair_hyrU · 2024-12-21

**Metareview:**

This paper addresses two key issues in current research on Large Language Model (LLM) tuning. First, it points out that parameter-efficient fine-tuning (PEFT) cannot fully compete with full-model tuning, as it only tunes a subset of model parameters, limiting the model’s capacity. Second, the paper identifies that zeroth-order (ZO) optimization can tune all model parameters but relies on fixed, non-optimal prompts, which may not be ideal. To address these challenges, the authors propose complementing ZO with PEFT through a bi-level optimization approach to reduce sensitivity to hard prompts. In their formulation, they first transform the problem into a single-level optimization problem and then use zeroth-order information to approximate the gradient.

However, the experimental section lacks clarity and efficiency comparisons. The advantage over the MeZO baseline is not sufficiently explained, and the authors do not address other baseline methods that could improve MeZO. Furthermore, the motivation for combining ZO with PEFT is not fully articulated. While bi-level optimization is known for its high computational cost, even with zeroth-order approximations, the combination of these methods appears relatively simple. Overall, the motivation behind this paper remains unclear.

**Additional Comments On Reviewer Discussion:**

The authors have addressed some of the reviewers' concerns. However, as pointed out by the reviewers, it remains unconvincing that MeZO suffers from suboptimal gradient estimation. Furthermore, it is still unclear why combining MeZO with PEFT would only lead to a decrease in performance. The improvement over the baselines is not substantial, and as a result, the reviewers have maintained their original rating.

---

### Decision · Program_Chairs · 2025-01-22

Reject